# The origin and structural evolution of de novo genes in *Drosophila*

Junhui Peng [ID][1] & Li Zhao [ID][1] ✉

Recent studies reveal that de novo gene origination from previously non-genic sequences is a common mechanism for gene innovation. These young genes provide an opportunity to study the structural and functional origins of proteins. Here, we combine high-quality base-level whole-genome alignments and computational structural modeling to study the origination, evolution, and protein structures of lineage-specific de novo genes. We identify 555 de novo gene candidates in *D. melanogaster* that originated within the *Drosophilinae* lineage. Sequence composition, evolutionary rates, and expression patterns indicate possible gradual functional or adaptive shifts with their gene ages. Surprisingly, we find little overall protein structural changes in candidates from the *Drosophilinae* lineage. We identify several candidates with potentially well-folded protein structures. Ancestral sequence reconstruction analysis reveals that most potentially well-folded candidates are often born well-folded. Single-cell RNA-seq analysis in testis shows that although most de novo gene candidates are enriched in spermatocytes, several young candidates are biased towards the early spermatogenesis stage, indicating potentially important but less emphasized roles of early germline cells in the de novo gene origination in testis. This study provides a systematic overview of the origin, evolution, and protein structural changes of *Drosophilinae*-specific de novo genes.

De novo genes are novel genes born from scratch from previously non-genic DNA sequences[1–3]. Recent works support the existence of a considerable number of young de novo genes across various species and taxa, including humans and *Drosophila*[4–14]. While some studies proposed that proteins encoded by de novo genes tend to be highly disordered[15] to prevent misfolding or aggregation, which can be neurotoxic in complex eukaryotes[15], other studies suggest that de novo genes may not necessarily be disordered[16–18] and instead suggest that their structures could be highly conserved after their origination.

Despite the aforementioned advancements, our understanding of the protein structures of de novo genes remains very limited. It is still unclear whether de novo genes are capable of being well-folded, how frequently they are well-folded, and if they possess novel structural folds. The main obstacle has been the lack of accurate, efficient, and scalable structural characterization tools that could be applied to a

large amount of de novo genes. Here, we applied AlphaFold2[19] computational predictions as well as ESMFold[20] to evaluate the foldability of de novo genes. With the rapid development in genome sequencing, sequence alignment, and deep learning techniques, AlphaFold2, along with other neural network approaches, e.g., trRosetta[21], RoseTTAFold[22], and a language model approach ESMFold[20], have demonstrated the ability to predict protein structures with near-atomic accuracy. AlphaFold2 has been applied at genomic scales to predict protein structures of the human proteome and the proteomes of several other species[23,24]. Although AlphaFold2 has been proven to be highly accurate, it predicts only a single static protein structure per protein sequence[25], which could hinder our understanding of the protein structures of de novo genes since proteins can be highly dynamic in cells. Molecular dynamics (MD) simulation has shown to be a valuable tool to investigate protein dynamics[26], study protein

---

[1]Laboratory of Evolutionary Genetics and Genomics, The Rockefeller University, New York, NY, USA. ✉e-mail: lzhao@rockefeller.edu

structure stability[27], and evaluate or refine predicted or designed protein structures[28,29]. Thus, we further carried out large-scale MD simulations to characterize the structural stability and dynamics of the predicted protein structures. In addition, the increasing power of bioinformatic and computational approaches has made it possible to obtain highly accurate general structural properties of proteins, including intrinsic structural disorder[30], relative solvent accessibility[31], and the probability of being transmembrane proteins[31] or containing a signal peptide[32].

In addition to the protein structures of de novo genes being unknown, the evolution of their sequences and protein structures after origination also remains unclear. To address this question, it is necessary to identify and compare branch-specific de novo genes of varying ages within a relatively diverged lineage. However, due to low genome sequencing quality and high genome recombination rates in some species, this process can be very difficult, especially with increased divergence time[4,33]. This limitation greatly hampers our understanding of the origin of de novo genes and how their sequences and protein structures evolve after origination. In addition, it has been historically difficult to distinguish between rapidly evolving genes and de novo originated genes[34,35]. However, with the recent advancement in whole-genome sequence alignments and its ability to progressively align the genomes of an entire phylogenetic tree, including diverged species with high accuracy[36], we can now identify de novo protein-coding gene candidates with high confidence through the support of synteny-based alignments and non-coding sequences in outgroup species.

In this work, we utilize progressive whole-genome alignments and multiple homology detection methods to identify 555 de novo protein-coding gene candidates in *D. melanogaster* that were born within the last ~67 million years since the *Drosophilinae* lineage[37] with the support

of orthologous non-coding DNA sequences in their corresponding outgroup species. We performed bioinformatic analysis and computational structural modeling to predict the structural and functional properties of each de novo gene candidate. We observe a gradual shift in sequence composition, evolutionary rates, and expression patterns with their gene ages, which indicates possible gradual shifts or adaptation of their functions. Surprisingly, there are few overall protein structural changes for de novo genes in the *Drosophilinae* lineage. We identify a number of de novo gene candidates with protein products that are potentially well-folded. Interestingly, ancestral sequence reconstruction (ASR) analysis reveals that most well-folded candidates are also well-folded in their ancestral stages. In addition, we observe one case where disordered ancestral proteins became ordered within a relatively short evolutionary time. Our results provide a systematic overview of the foldability of proteins encoded by de novo genes and the patterns in which their sequences and structures evolve after origination.

## Results

### Identification of *Drosophilinae* lineage-specific de novo gene candidates in *D. melanogaster*

We built the whole genome alignment of 20 *Acalyptratae* fly species (Fig. 1a) using Progressive Cactus Aligner[36]. A summary of the alignments can be found in Fig. S1. Overall, the protein-coding bases in *D. melanogaster* can be aligned to closely related species at high coverage (97% with *D. sechellia*), while the coverage dropped significantly to around 50% in relatively distant species (49% with *B. dorsalis*). For the 13968 annotated protein-coding genes in *D. melanogaster* investigated in our study, we observed that 13798 (98.8%) of them were covered in Cactus alignments. For each of the 13798 *D. melanogaster* protein-coding genes, we combined homology obtained from all-vs-all blastp[38]

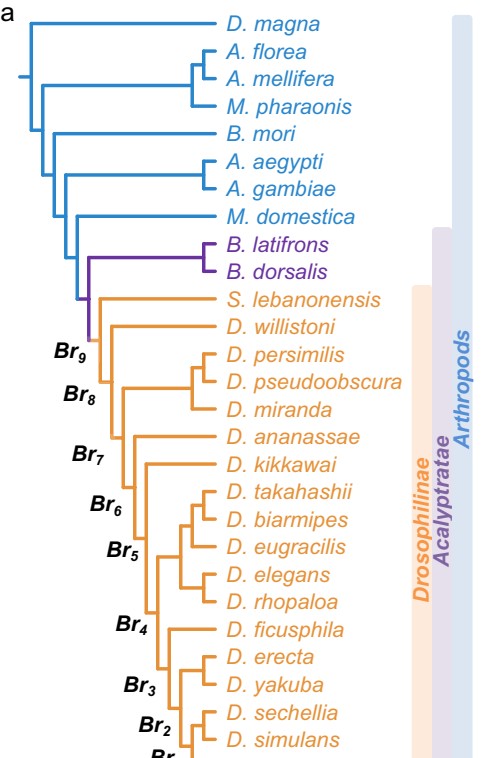
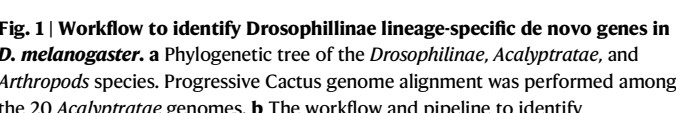

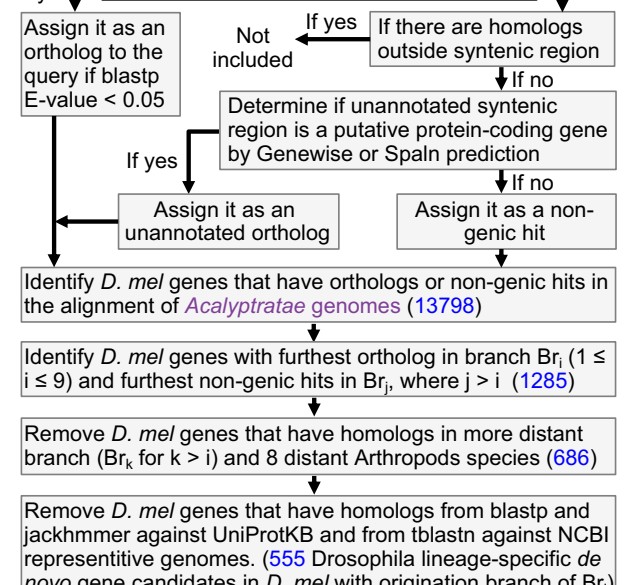

**Fig. 1 | Workflow to identify Drosophillinae lineage-specific de novo genes in *D. melanogaster*. a** Phylogenetic tree of the *Drosophilinae*, *Acalyptratae*, and *Arthropods* species. Progressive Cactus genome alignment was performed among the 20 *Acalyptratae* genomes. **b** The workflow and pipeline to identify *Drosophilinae* lineage-specific de novo genes and track the origination within the *Drosophilinae* lineage for the current protein-coding genes in *D. melanogaster* (see *Methods*). The numbers of potential candidates in each identification step were highlighted in blue.

analysis, Genewise[39] and Spaln[40] predictions to identify annotated/unannotated orthologs and non-genic hits from their syntenic regions (Fig. 1b). For simplicity, we termed the furthest branches that have annotated/unannotated orthologs as Br$_i$, where i could range from 1 to 9 for each potential candidate, as shown in Fig. 1a. The above step gave 1285 potential de novo gene candidates within Br$_9$ (see Fig. 1b, *Methods*, and Supplementary Data 1 for details). We then removed genes that have homologs that are not in the syntenic regions using all-vs-all blastp[38]. This led to 686 potential de novo gene candidates within Br$_9$. As the last filtering step, we removed candidates that have reliable annotated or unannotated homologs outside of Br$_i$ by blastp and iterative jackhmmer[41] search against UniProt Knowledgebase sequence database (UniprotKB)[42] and tblastn search against NCBI representative genomes (Fig. 1b, *Methods*). Finally, combined with homology and synteny, we identified 555 de novo protein-coding gene candidates in *D. melanogaster* that potentially originated within *Drosophilinae* lineage (Supplementary Data 2). Of these genes, 397 were born from intergenic regions and 158 from intragenic regions. All these de novo gene candidates were supported by evidence of possible ancestral non-coding DNA sequences. For each of the branches (Br$_1$, Br$_2$ to Br$_9$), the number of de novo gene candidates originated in the branch is shown in Fig. S2. We did not observe de novo genes that are *D. melanogaster* specific due to the identification of putative unannotated orthologs (see *Methods*) in unannotated syntenic regions in the outgroups. However, we did observe 73 *D. melanogaster* protein-coding genes with unannotated syntenic regions in *D. simulans, D. sechellia*, or other species in more distant branches. Of these 73 genes, 50 were not considered to be de novo genes since the unannotated syntenic regions were predicted to be unannotated orthologs (see *Methods*) with high confidence. The remaining 23 genes were identified as de novo gene candidates, but were inferred to be originated from more distant branches.

## The origin of *Drosophilinae*-specific de novo genes is more likely to be associated with open chromatin than transposable elements

Our analysis shows that most de novo gene candidates had biased expression in the testis, head, and ovary. Thus, to understand their relationships with open chromatin regions, which are enriched with regulatory sequences, we examined ATAC-seq data from the three tissues[43]. We found that both intergenic and intragenic de novo gene candidates have more peaks in their nearby ± 500 bp regions than putative random ORFs in intergenic regions. In these regions, the peaks of the de novo gene candidates also have higher peak intensities. When considering broader regions, we found that de novo gene candidates, regardless of being intergenic or intragenic, were closer to their nearest peaks. This suggests that de novo genes are partly associated with open chromatin conformation changes. To assess the potential involvement of TEs in de novo gene origination, we searched for DNA repeat signals (see *Methods*) in these regions. However, we did not find evidence of an association between de novo gene candidates and TEs in *Drosophila*, contrasting with previous reports of up to 20% of de novo transcripts being associated with TEs in primates, including humans[44]. One possible explanation is that the primate genomes have much higher TE contents than fruit flies[45]; thus, the potential regulatory roles of TEs in *Drosophila* new genes are limited. Our findings suggest that in *Drosophila*, lineage-specific de novo genes are associated with open chromatin regions (Fig. S3) but not TEs, a different pattern from what has been observed in primates.

## De novo gene candidates are mostly adaptive and shaped by both adaptive and non-adaptive changes

Compared to other annotated genes, de novo gene candidates display distinct properties in sequence composition (GC content), sequence evolution ($\omega$, $\omega_a$, $\omega_{na}$, and $\alpha$), structural properties, and expression patterns (Fig. 2). Specifically, these candidates exhibit lower GC contents (Fig. 2a), and this trend applies to the codons of each amino acid that contains G or C in their codons (Fig. 2b). We further computed the optimized codons for each amino acid within the *D. melanogaster* genome (see *Methods*). We found that de novo genes tend to use less optimized codons than other protein-coding genes (Table S1). These findings suggest that de novo genes might be using unoptimized codons and that selection on codon usage might play an important role in their evolution.

Compared to other protein-coding genes, the protein products of de novo genes tend to be more disordered ($p = 2e-80$), more exposed ($p = 4e-80$), and are more likely to be transmembrane proteins ($p = 4e-6$) or secretory proteins ($p = 6e-11$) (Fig. 2c). De novo genes have higher male specificity ($p = 6e-84$), higher tissue specificity ($p = 3e-88$), relatively lower expression levels in females ($p = 5e-78$), and slightly but not significantly higher expression levels in males ($p = 0.09$). Many of these patterns are consistent with observations in studies that focused on less divergent species groups in *Drosophila* or other taxonomy[11–13,46,47], further indicating that de novo genes exhibit some universal patterns that are not dependent on their lineage or identification method.

By analyzing comparative and population genomics data, we found that de novo genes are under faster sequence evolution compared to other protein-coding genes (Fig. 2e). The adaptation rates, nonadaptation rates, and proportions of adaptive changes of de novo genes are higher than other genes (Fig. 2e), indicating that both adaptive evolution and relaxation of purifying selection contribute to the elevated evolutionary rates of de novo genes. The results that de novo genes are more likely to be transmembrane or secretory proteins (Fig. 2c) and male-specific or tissue-specific (Fig. 2e) suggest that some de novo genes might have specific molecular or cellular function[18,48,49].

## De novo gene candidates undergo gradual sequence/function changes without significant structural changes

To understand whether and how structures change after the fixation of a de novo protein-coding gene, we then studied structural differences in de novo genes among different origination branches or gene ages. We compared the differences in protein properties of de novo genes with origination branches (Br$_1$ to Br$_9$, Fig. 1a) by applying Kendall tau and Spearman's rank correlation analysis. We observed significant sequence changes among different origination branches. For example, the sequence evolutionary rates ($\omega$) of de novo genes are significantly negatively correlated with their respective origination branches (Kendall tau $P = 1e-11$, Spearman's rank $P = 5e-12$, Fig. 2f, i). The significant changes of $\omega$ might be dominated by the decrease of nonadaptive changes ($\omega_{na}$) since non-adaptation rates had significant negative correlations with their respective origination branches as well (Kendall tau $P = 5e-5$, Spearman's rank $P = 3e-5$, Fig. 2f, i), while the negative correlations between adaptation rates and their respective origination branches were not significant (Kendall tau $P = 0.9$, Spearman's rank $P = 0.9$, Fig. 2f, i). The decrease of non-adaptive changes ($\omega_{na}$) might indicate an increase in the strength of purifying selection, suggesting the functional importance of de novo genes increases over limited evolutionary time.

We found that GC contents are lower for de novo genes originated in younger branches and lower in older branches (Kendall tau $P = 4e-6$, Spearman's rank $P = 3e-6$, Fig. 2I). The same trend applies to the GC content of the codons of each amino acid (Table S1). We computed the optimal codon, the most frequently used codon, for each amino acid. We found that younger de novo genes used significantly less optimal codons than older de novo genes (Table S1). These observations might suggest an important role of selection on codon usage or translation in the evolution of de novo genes. We observed significant correlations between male and tissue specificity and origination branches (Kendall

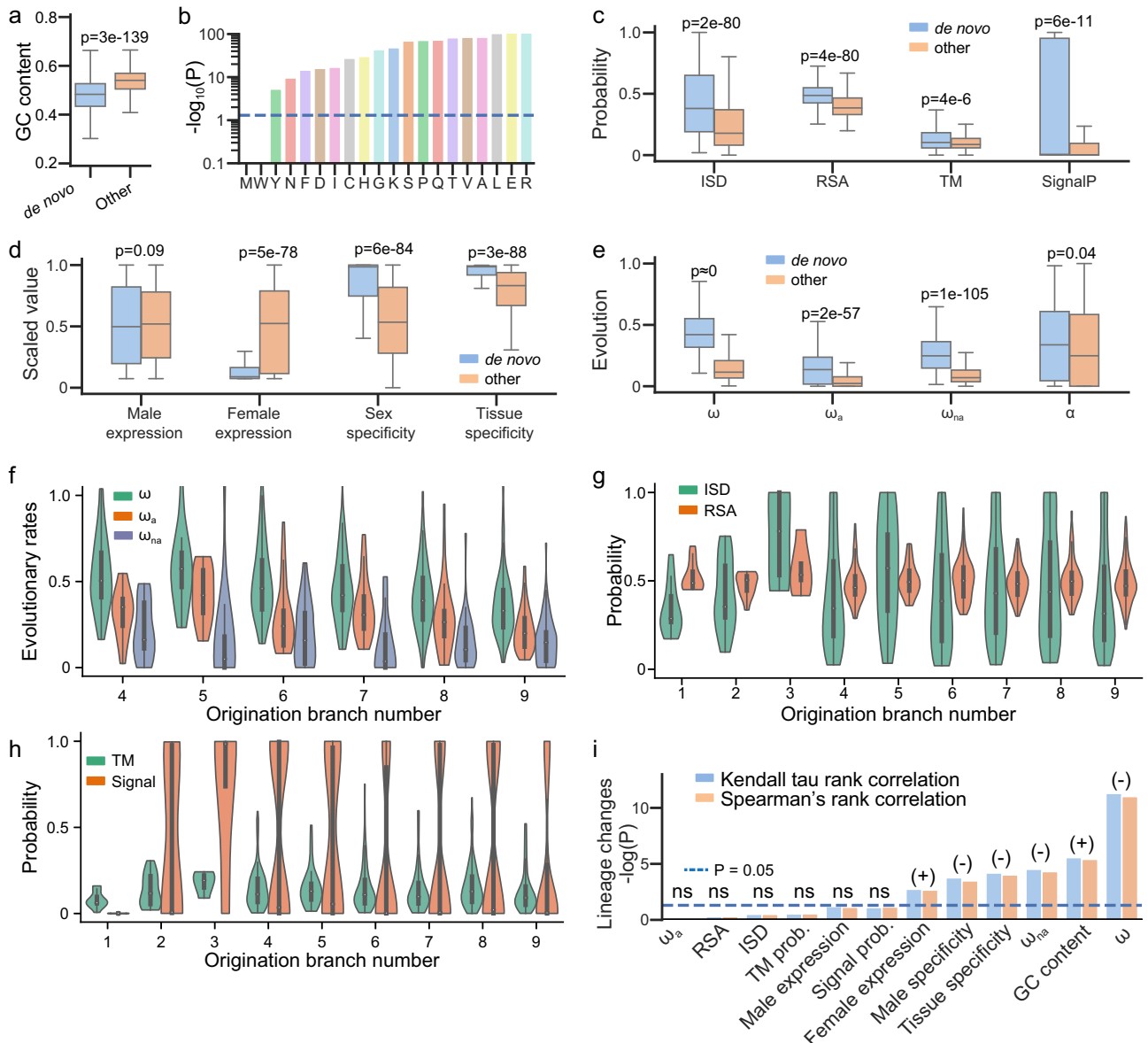

**Fig. 2 | Special properties of de novo gene candidates. a** GC contents of de novo gene candidates (denoted as "de novo") were significantly lower than other annotated protein-coding genes (denoted as "other"). **b** GC contents of the codons utilized by each of the amino acids in de novo gene candidates were significantly lower than those of the amino acids in other annotated protein-coding genes. De novo gene candidates also differed in **c** structural properties, **d** expression patterns, and **e** evolutionary patterns. For structural properties, structural disorder (ISD), solvent exposure (RSA), transmembrane probability (TM), and signal protein probability (SignalP) are shown. For evolutionary patterns, evolutionary rates (ω), adaptation rates (ω_a), nonadaptation rates (ω_na), and proportions of adaptive changes (α) are shown. **f** Evolutionary rates of de novo gene candidates with different gene ages. Due to insufficient data, Br_1, Br_2, and Br_3 were not shown. **g** Structural disorder (ISD) and solvent accessibility of de novo gene candidates

with different gene ages. **h** The probability of containing transmembrane or signal proteins for de novo gene candidates with different gene ages. **i** Correlations between different properties of de novo gene candidates and their gene ages. Positive, negative, and not significant correlations were indicated by (+), (−), and ns, respectively. Expression and evolutionary patterns of de novo gene candidates were significantly correlated with gene ages, while structural properties were not. P-values in **a**–**e** were examined for the 555 de novo gene candidates over other 13413 annotated protein-coding genes in *D. melanogaster* by one-sided t-test. Boxplots in **a** and **b**–**e** are plotted with whiskers extending to ±1.5 × IQR. P-values in **f** to **i** were all examined for the 555 de novo gene candidates over nine different origination branches, by two-sided Spearman or Kendall's tau rank tests, unless otherwise noted. Source data is provided in the source data file.

tau $P = 3e-4$ and $2e-4$, Spearman's rank $P = 1e-4$ and $7e-5$, Fig. 2i). Similarly, we observed weaker but significant correlations for female expression levels (Kendall tau $P = 2e-3$ and $2e-3$), and weaker but not significant correlations for male expression levels (Spearman's rank $P = 0.07$ and $0.06$, Fig. 2i). The correlations between expression patterns and origination branches might indicate a gradual change in protein functions, which further suggest that these de novo genes are under certain degrees of selection.

Interestingly, we did not observe significant changes in structural properties among different origination branches, including structural disorder (ISD), solvent accessibility (RSA), probability of being transmembrane proteins, and probability of being signal proteins from Kendall tau and Spearman's rank test (Fig. 2g–i, Fig. S4). This indicates that upon origination, the overall structural properties of de novo genes might remain similar in *Drosophilinae* lineage. The correlations between de novo gene properties and their gene ages indicate that

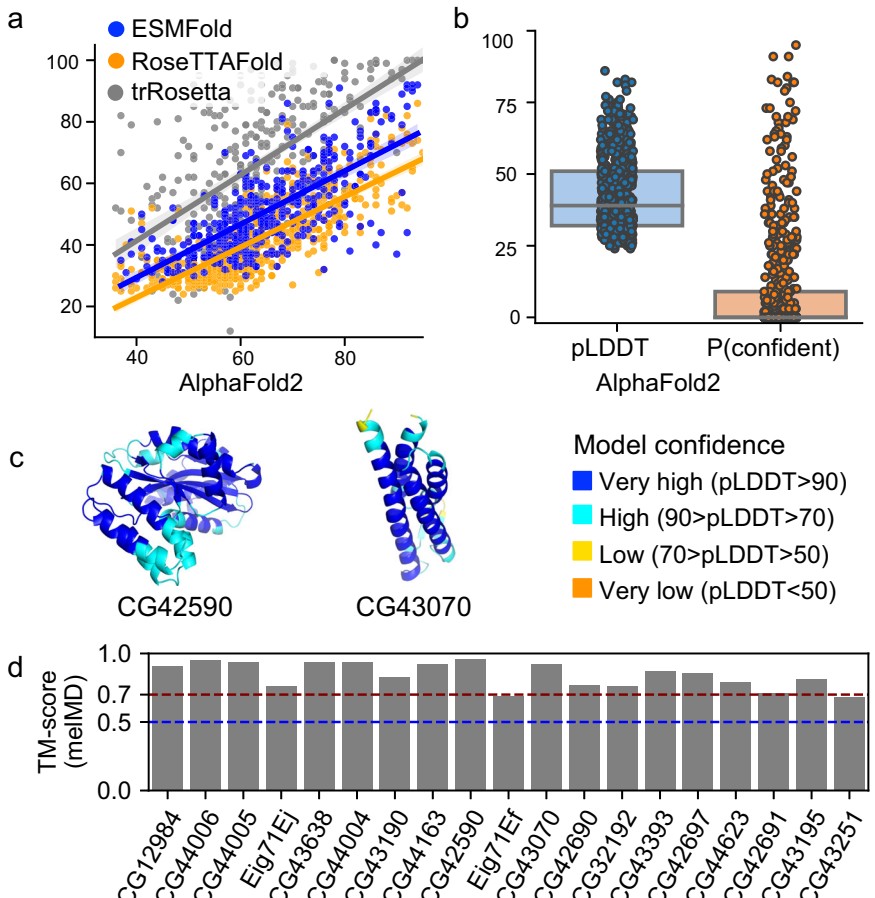

**Fig. 3 | AlphaFold2 structure predictions of de novo gene candidates.**
**a** AlphaFold2 predictions are strongly correlated with RoseTTAFold and trRosetta predictions. The confidence interval is estimated by 1000 bootstrap resamples and is shown in shadow. **b** Most de novo gene candidates might not have well-folded protein structures according to pLDDT, per-residue confidence score, and P(confident), which represents the percentage of residues that were confidently predicted by AlphaFold2 with pLDDT greater than 70. Boxplot ($n = 555$) is plotted with whiskers extending to $\pm 1.5 \times$ IQR. **c** Two examples of potentially well-folded de novo gene candidates: CG42590 and CG43070. In both cases, the core regions were predicted with very high confidence (blue). **d** The predicted structural models of the 19 potentially well-folded de novo genes retained highly similar structural folds during MD simulations with pairwise TM-score of representative structures close to or larger than 0.70. Source data is provided in the source data file.

these de novo gene candidates might undergo gradual sequence and functional changes without significant structural changes.

## A small subset of de novo genes is potentially well-folded with complex structural folds

We used AlphaFold2[19] to predict the structural models of de novo gene candidates. We further used the per-residue confidence score (pLDDT) from AlphaFold2 predictions to estimate the foldability of these candidates as high pLDDT scores often indicate accurate protein folding, and low pLDDT scores highly correlate with protein disorder[19]. We showed that pLDDT from AlphaFold2 predictions strongly correlates with pLDDT from RoseTTAFold[22] predictions (Pearson correlation R = 0.77, P = 1e-43), ESMFold[20] predictions (Pearson correlation R = 0.76, P = 2e-42), and the convergence of trRosetta[21] predictions (Pearson correlation R = 0.60, P = 2e-22) (Fig. 3a, see *Methods*), suggesting consistency among the three state-of-the-art prediction methods. We categorized de novo gene candidates into three structural groups, as either (1) potentially well-folded, (2) partially folded, or (3) not folded. We defined a gene to be potentially well folded if its average per-residue confidence score (pLDDT) was greater than 80 and the percentage of confidently predicted residues (pLDDT > 70) greater than 90%; a gene to be potentially partially folded if more than 30% of its residues or more than 50 consecutive residues being confidently predicted (pLDDT > 70); and the remaining genes to be potentially not folded. We found that most of the de novo gene candidates might only be partially folded (297/555) or not folded (224/555) (Fig. 3b, Supplementary Data 2). Interestingly, we found 34 de novo gene candidates that are potentially well-folded. Among these 34 de novo genes, 16 might only fold into simple folds containing one or two α-helices, while another 19 might fold into complex folds (Table S2). We further performed three independent 200 ns MD simulations for each of the 19 structural models to refine the structural models and characterize the stabilities of the structural folds (see *Methods*, Table S3). The structural models all retained highly similar structural folds during the MD simulations, with pairwise TM-scores of representative conformations close to or larger than 0.70 (Fig. 3d). For example, CG42590 and CG43070 had averaged pLDDT scores of 90 and 89 (Fig. 3c, Table S2). Their structural folds remained highly stable during MD simulations with averaged pairwise TM-score values of 0.96 and 0.92 (Table S3).

## Most potentially well-folded de novo genes adopt existing protein structure folds

To check if the 19 potentially well-folded de novo genes have novel structural folds, we compared their MD-refined structures with all experimentally determined protein structures in Protein Data Bank (PDB)[50]. We did this by searching against PDB for potential novel structural folds using RUPEE[51]. Interestingly, we found that most of

them (16/19) have similar structural folds in PDB (Table S2). These similar structural folds were not due to sequence similarity. In contrast, the sequence identities between the de novo gene candidates and their similar structures were mostly less than 10% (Table S2). For example, the structural model of CG43195 is similar to the A chain of PDB structure 1U89 with a TM-score of 0.70. However, their sequence identity is only 4% (Table S2). Notably, we also found 3 de novo gene candidates, Eig71Ei, Eig71Ej, and CG43251, with maximum TM-score against protein structures in PDB smaller than 0.5 (Table S2, Fig. S5), suggesting that the 3 candidates might adopt novel structural folds that have not been identified before. Overall, our results indicated that many of the potentially well-folded de novo proteins examined in our study are likely to adopt existing protein structure folds.

## Most potentially well-folded de novo genes are likely to be born well-folded

We showed that de novo gene candidates could undergo fast sequence adaptation without significant structural changes. We further investigate the structural changes of de novo genes after origination using

the 19 potentially well-folded de novo genes as examples. For all the 19 de novo genes, we reconstructed their ancestral states, including the most ancestral states and intermediate states in other branches, by ancestral sequence reconstruction (ASR). We then used AlphaFold2 to predict the 3D structures of these ancestral states. We found that the ancestral states of the 19 de novo genes were all predicted to be potentially folded at high confidence (average pLDDT > 70, Table S2). We further compared the structural models of ancestral states to current states. We found that they share similar structural folds with all pairwise TM-scores greater than 0.56 (Fig. 4a, top panel), suggesting that these potentially well-folded de novo genes are likely to be born with similar structural folds to the current forms. To refine the structural models and characterize the stabilities of the structural folds, we further conducted three independent 200 ns MD simulations starting from each of the ancestral structural models (see *Materials and Methods*, summarized in Supplementary Data 3). We found that most of the ancestral state structural models retained similar structural folds to their current *D. melanogaster* forms during MD simulations, with almost all pairwise TM-scores of representative conformations close to

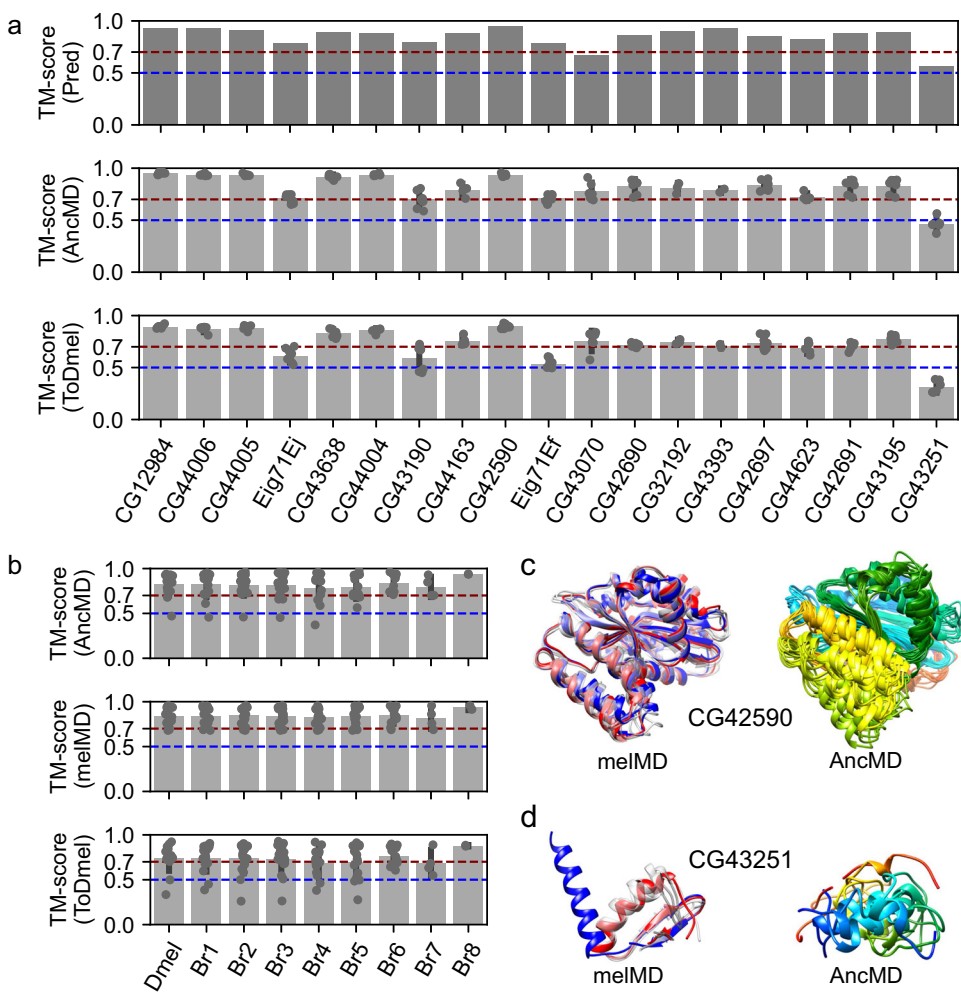

**Fig. 4 | Structural evolution of 19 potentially well-folded de novo gene candidates. a** Ancestral states have similar structural folds to their current forms, as predicted by AlphaFold2 ("Pred", top panel). The structural folds of ancestral structural models were stable ("AncMD", middle panel) and similar to their current forms ("ToDmel", bottom panel) during MD simulations, except for the case of CG43251 (last column). **b** Structural fold stability of ancestral states ("AncMD", top panel), and current forms ("melMD", middle panel) as a function of the ancestral state ages. Structural models of ancestral states were similar to their current forms, regardless of their ancestral branches ("ToDmel", bottom panel). In the middle and

bottom panels of **a** and all panels of **b**, data are presented as mean values±SD. **c, d** Two examples of the structural evolution of potentially well-folded de novo gene candidates. AlphaFold2 predicted structural model (blue), MD refined structural model (red), and other representative structural models (gray) of CG42590 (**c**, left panel) and CG43251 (**d**, left panel). MD refined structural models of different ancestral states of CG42590 (**c**, right panel) and CG43251 (**d**, right panel) colored by residue index, with the N-terminus being blue and C-terminus red. Ancestral states of CG42590 have similar structural folds, while those of GC43251 are different. Source data is provided in the source data file.

or larger than 0.70 (Fig. 4a, middle panel and bottom panel). In addition, we found that the structural fold stabilities and their similarities to their current forms did not change with the age of ancestral states (Fig. 4b). Taking CG42590 as an example, MD simulations of CG42590 in its current form revealed that it contains a highly stable structural fold (Fig. 4c, left column). Meanwhile, MD refined structural models of CG42590 ancestral states revealed that they are highly similar to CG42590 current form (Fig. 4d, right column), with an average TM-score of 0.90. Interestingly, we also observed one exception, CG43251. We found that CG43251 might have undergone substantial global structural changes from ancestral states to its current state (Fig. 4a, last column), with an average TM-score of 0.31 to its current form. In the MD refined structure of the current form, CG43251 forms a stable fold with an alpha-helix in the N-terminal and a β-hairpin in the C-terminal (Fig. 4d, left column), while in its ancestral states, the structures were disordered (Fig. 4d, right column). Altogether, the results suggested

that after origination, most potentially well-folded de novo genes might preserve a similar fold as they were born, while in some rare cases, the candidates undergo substantial structural changes.

## Early germline cells in the testis are non-negligible in de novo gene origination, despite that most de novo genes are enriched in later germ cells

Of the 555 de novo gene candidates identified, many of them (217, ~40%) had biased expression in the testis. To investigate how de novo genes originated and evolved in testis, we analyzed the expression patterns of testis-biased de novo genes using testis single-cell RNA-sequencing data[52]. We used the expression patterns of all annotated testis-biased *D. melanogaster* genes to cluster these genes into four different clusters (Fig. 5a, Fig. S6). We numbered the four clusters according to the expression patterns, where genes in cluster #1 tend to be highly expressed in early spermatogenesis stage, genes in cluster #2

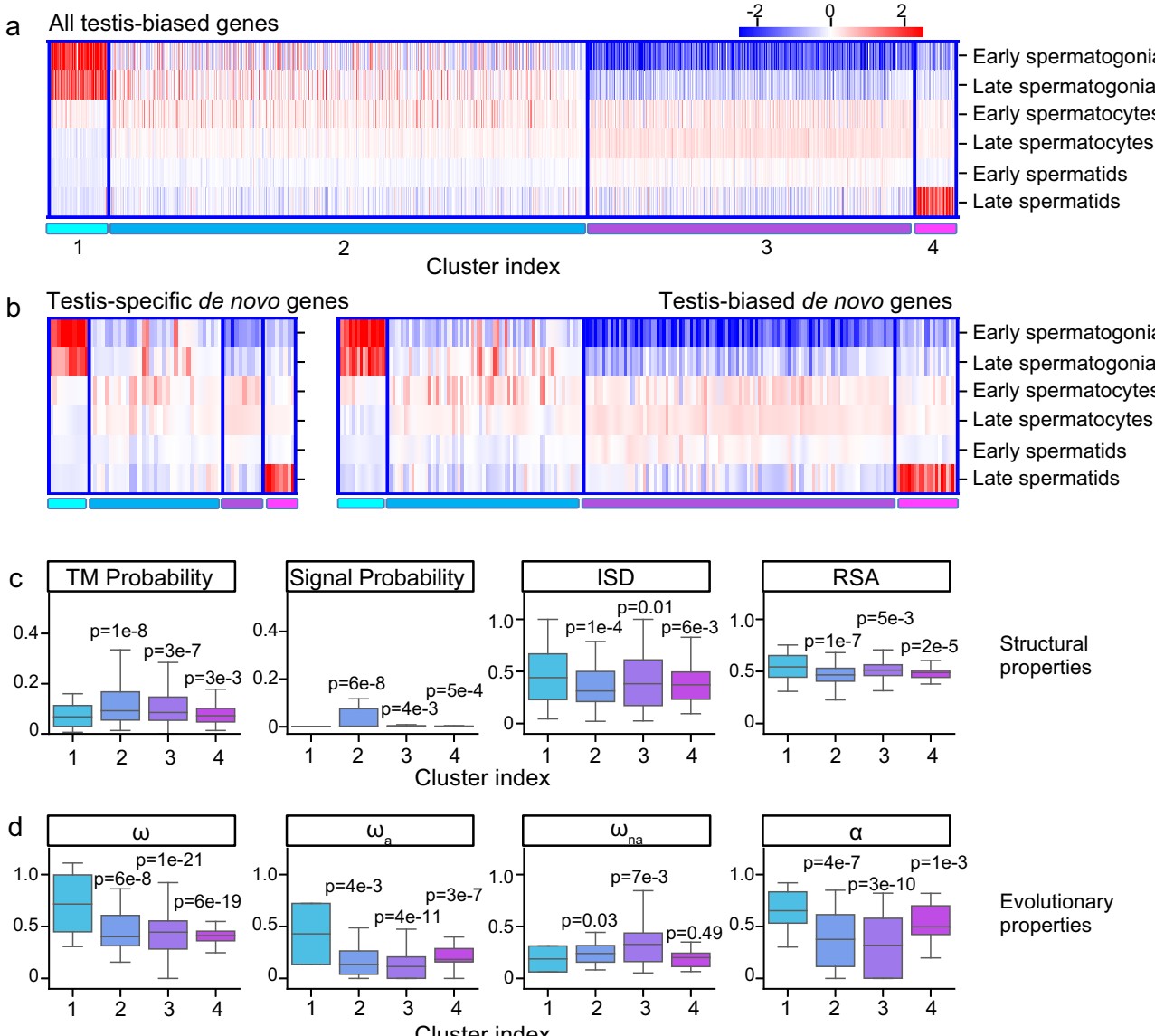

**Fig. 5 | Expression pattern of testis-biased de novo gene candidates. a** Clustering of all *D. melanogaster* testis-biased genes. **b** Expression patterns of testis-specific (left) and testis-biased (right) de novo gene candidates in different clusters. **c** de novo gene candidates in Cluster #1 are less likely to be transmembrane (TM probability) or contain a signal peptide (Signal probability) but are more likely to have higher structural disorder (ISD) and higher solvent accessibility (RSA). **d** de novo gene candidates in Cluster #1 have higher evolutionary rates ($\omega$), adaptation rates ($\omega_a$), and proportions of adaptive changes ($\alpha$), but not higher nonadaptation rates ($\omega_{na}$). *P*-values in **c**, **d** were examined for de novo gene candidates in cluster #1 ($n = 22$) against candidates in other clusters ($n = 70$ for cluster #2, 106 for cluster #3, and 23 for cluster #4) by one-sided t-test. Boxplots are plotted with whiskers extending to ±1.5 × IQR. Source data is provided in the source data file.

showed average expression in spermatogonia and spermatocytes stages, genes in cluster #3 showed average expression in spermatocytes and spermatids stages, and genes in cluster #4 showed peak expression in spermatids stage. We found that the testis-biased de novo genes' expression profile is similar to other testis-biased genes (Fig. 5b). We then asked whether de novo genes in different clusters showed different structure/sequence characteristics. We found that de novo gene candidates in cluster #1 were significantly different from other genes. For example, these de novo gene candidates were less likely to be transmembrane or secretory proteins, and they were more likely to be disordered or exposed proteins than de novo gene candidates in other clusters (Fig. 5c). We also found that, compared to other clusters, de novo genes in cluster #1 evolve faster, and the faster evolutionary rates were mostly contributed by their faster adaptation rates (Fig. 5d). These special properties of de novo genes in cluster #1 were not because testis-biased genes in cluster #1 have these properties. We compared de novo gene candidates and other testis-biased genes in cluster #1. We found that in cluster #1, de novo proteins were more likely to be disordered and exposed than other testis-biased genes, and that de novo genes tend to evolve faster with faster adaptation rates. In cluster #1, the faster evolutionary rates of de novo genes were also more likely to be attributed to their faster adaptation rates (Fig. S7). While previous studies emphasized the importance of mid-to-late spermatogenesis stages in de novo gene origination[4,52–54], our results indicate that although only very few de novo genes were enriched in the early spermatogenesis stages, they might also play a non-negligible role in de novo gene origination.

## Discussion

In the past decade, various synteny-based methods have been applied to determine the de novo emergence of de novo genes[4]. However, many of them suffered from divergence between genomes in longer timescales, fragmented genome assemblies, and high genome recombination rates[4]. Thus, in this study, we used whole-genome alignments instead of just synteny mapping to determine the ortholog regions of *D. melanogaster* protein-coding genes. Specifically, we used the base-level progressive Cactus aligner, which has been proven to be highly accurate in both pairwise genome alignments[55] and progressive genome alignments of an entire phylogenetic tree[36]. Further analysis showed that Cactus aligner recovered more syntenic regions and orthologs compared to a micro-synteny method (section *Micro-synteny and orthoMCL analysis* in *Methods*, Fig. S8, Table S4), indicating the advantage of Cactus aligner over synteny methods in de novo gene identification. With the high-quality base-level whole genome alignments, we can identify de novo genes with the support of alignments against non-coding sequences in outgroup species. The aligned non-coding sequences could also facilitate the discovery of possible origination events of de novo genes. Despite these merits, there are still some inherent drawbacks to this method. The first is that we did not include taxonomically restricted genes (TRG) or orphan genes that cannot be aligned in progressive Cactus alignments, which could underestimate the number of true de novo originated genes. However, it is also important to note that, although TRG or orphan genes are likely to be de novo emerged, without the support of non-coding alignments, we cannot rule out the possibility of other mechanisms, such as horizontal gene transfer (HGT), transposition, etc. The second drawback is that we used simulated data to distinguish unannotated orthologs and unannotated non-coding regions. For genes with unannotated syntenic regions, we define orthology if their gene structure prediction scores were significantly beyond random expectations. In this case, unannotated ortholog regions with frameshift indels but high prediction scores might also be considered orthologs. Thus, it is likely we assign a number of de novo genes to a more ancestral branch than their true origin dates. However, we argue that, according to the high average genomic mutation rates in *Drosophila*

species[56], an unannotated ortholog region with high prediction scores might be under strong purifying selection or might just be dead recently, otherwise this region would be completely random after random neutral mutations for millions of years. This compromise might be necessary as the misidentification of old genes as young genes may lead to biased patterns[34,35,57]. Similarly, de novo genes from older branches may still suffer more from both false positives and false negatives. We further found gene expression evidence from five different *Drosophila* species for many of these unannotated regions (Fig. S9). Note that it is still likely that we might have excluded potential genic sequences that were too diverged to have meaningful gene structure predictions[40]. Another drawback was that the estimation of origination branch might be affected by the unavailability of high-quality genomic data for some other species in the branch of interest. With the great efforts being made to improve the genome sequencing and annotation qualities, we will be able to better study de novo gene emergence events in the future.

There were a few case studies trying to characterize the foldability and structure of proteins encoded by some specific de novo originated genes[16–18]. However, it remains unclear whether de novo genes could be well-folded with complex structural folds, how often de novo genes could be well-folded, and whether de novo genes could have novel structural folds. Here, we explored the foldability of de novo genes/proteins by highly accurate and efficient AlphaFold2 predictions. In general, the AlphaFold2 predictions correlated significantly with other state-of-the-art neural network or language model approaches, such as trRosetta, RoseTTAFold, and ESMFold. It was also reported that AlphaFold2 predictions were comparable to high-resolution structure determination techniques, such as solution NMR and x-ray crystallography, especially for some small and relatively rigid single-domain proteins[58]. We found that most de novo genes might be partially folded or not folded, and only very few of them have the potential to be well-folded with complex structural folds (19/555), among which several (3/19) might have novel structural folds. The complex structural folds of these de novo gene candidates remained stable during three independent 200 ns MD simulations (Fig. 3d), which further suggested the high accuracy of AlphaFold2 predictions. Our results provided a systematic overview of the foldability of *Drosophilinae* lineage-specific de novo genes and highlighted the possibility that de novo proteins can be well-folded and even folded into novel protein folds. Note that, recent studies have shown that AphaFold2 may fold some small lineage-specific or de novo proteins into unrealistic simple low energy conformations[59–61]. In our study, we applied different deep learning predictors (AlphaFold2 and ESMFold) as well as MD simulations to partially overcome this limitation. However, since these observations were based on computational predictions, further experimental validation is needed to better understand the protein structures of de novo genes.

The fact that de novo genes were preferentially expressed in the testis has led to the hypothesis that de novo genes were "out-of-testis". The "out-of-testis" hypothesis was thought to be related to the facilitating role of the permissive chromatin state in germline cells[62]. Similar findings were observed in our study - we found that de novo genes were associated with open chromatin in the testis (Fig. S3), and that about half of the identified de novo gene candidates were biased toward the testis. These findings highlight the important role of testis and chromatin states in the origin of de novo genes. In addition, an examination of another two hotspot tissues, the head and ovary, suggested that de novo gene candidates were also associated with open chromatin in these two tissues (Fig. S3), highlighting the non-neglectable role of open chromatin in the emergence of de novo genes. In agreement with previous findings that spermatocytes and spermatids were the hotspots for de novo genes[52], we found that most testis-biased de novo gene candidates identified here have their peak expression in late spermatogenesis

stages, including spermatocytes and spermatids. The results might indicate that de novo genes that originated in late spermatogenesis stages may have higher probabilities of being fixed due to a stronger fitness effect in sperm competition, while those that originated in early spermatogenesis stages were more likely to be lost during evolution. On the other hand, we found that testis-biased de novo genes in early spermatogenesis stages showed some special properties compared to testis-biased de novo genes in later stages. For example, these genes tend to be more disordered or exposed and evolve faster with higher adaptation rates. Our results indicate a non-negligible role of early spermatogenesis stages in de novo gene origination. To further address the precise mechanisms, it would be necessary to conduct functional assays to examine the fitness effect of young and old de novo genes, which is beyond the current scope of this study.

It has been debated whether de novo genes change their protein structures after origination, and it is still unclear how de novo genes evolve within a limited evolutionary time scale. Some studies found that protein disorder increases with the age of de novo genes in *Lachancea*[47], and some other studies suggested de novo genes evolved or adapted gradually to avoid disorder[15]. While some case studies[16,18] and a recent study focusing on de novo ORFs[63] suggested that protein disorder and other properties hardly change after origination. Here, by comparing de novo genes originated from different branches, our results supported that protein disorder and other structural properties change little after origin. Interestingly, other properties, such as GC content, evolutionary rate, and expression pattern change with the age of de novo genes. Specifically, compared to younger de novo genes, older de novo genes have higher GC content, slower evolutionary rate, and are more broadly expressed in various tissues. The results also indicated that GC content was not necessarily the cause of high protein disorder in de novo genes, as protein disorder hardly changes while GC content decreases with the age of de novo genes. The changes in GC content might be contributed by mechanisms such as the gradual optimization of codons, which could potentially increase the translational efficiency of de novo genes. We noticed that de novo genes have increased strength of purifying selection (decreased $\omega_{na}$) with de novo gene ages, while the adaptation rates change little, which suggests that older de novo genes might be related to more important functions. In agreement with this result, it was reported that, at a larger evolutionary timescale, the number of protein-protein interactions (PPIs) and genetic interactions increased gradually with the age of genes[64]. Taken together, we propose that, upon origination, de novo genes were involved in some molecular functions with certain fitness effects, partly due to the tendency to encode signal proteins or transmembrane proteins[18,48,49]. In cases where de novo genes lost their functions, they might be depleted shortly after origination[10,46]. They tend to be in the periphery of cellular networks[64], and thus more likely to be tolerated by the host. Due to the weaker selective constraints, de novo genes tend to undergo faster sequence evolution, resulting in abundant sequence changes. These changes could potentially happen in regulatory or coding regions and further affect the expression patterns or expression levels of de novo genes. The gradual shift of sequence and expression patterns of de novo genes might increase their chances of being involved in protein-protein interactions (PPI), genetic interactions or other intermolecular interactions through adaptive evolution[65], which can further facilitate the integration of these genes into cellular networks. On the other hand, de novo genes might retain certain basic molecular functions by constraining similar overall structures (see section *Most potentially well-folded* de novo *genes are likely to be born well-folded* in Results) or key sequence motifs along their evolutionary trajectories.

Our study presents a comprehensive analysis of de novo genes of various ages, offering a systematic overview of their origination and evolution. While previous investigations in different lineages have employed diverse methodologies[10–14,34,44,47,52,62,66,67], several characteristics of de novo genes appear to be consistent across multiple lineages. Notably, de novo genes tend to be relatively short in length and exhibit a strong enrichment in the testis, displaying biased functions associated with this reproductive organ. Additionally, in mammals, the expression of de novo genes in the brain is relatively common[44,67,68]. This is also observed in older de novo genes in *Drosophila*, although to a much lesser degree. Intriguingly, our research highlights the potential for the immune system (GO enrichment *P*-value = 7e-4) to serve as another hotspot for de novo gene origination. However, further investigations are required to determine if this pattern holds true in other taxa or lineages. Furthermore, our investigation reveals that de novo genes predominantly exhibit a disordered nature, and this characteristic remains stable over the time frame examined. These findings align with a recent study[63]. In contrast, there are some inconsistencies in previous studies. For example, some studies observed in yeast that protein structural disorder increased with gene age[66], while others observed in yeast and mice that protein structural disorder decreased with gene age[15]. Investigating whether this discrepancy arises from methodological differences or possesses biological relevance warrants further exploration. For example, a recent study revealed that the choice of protein structural disorder predictors could result in discrepancies[59]. Our work is the first study to reveal structural conservation for well-folded de novo proteins using high-accuracy 3D structure modeling; whether this is a conserved pattern awaits future studies in other taxa or lineages. Another major point of contention revolves around the origin of de novo genes: whether they arise through neutral processes or are driven by strong selection. Our study on *D. melanogaster*, a species with a large effective population size, demonstrates that both adaptive and nonadaptive changes play pivotal roles in the slightly accelerated evolution of de novo genes after their birth. Previous studies that have distinguished adaptive and nonadaptive rates in de novo gene evolution are scarce. Exploring the applicability of our findings to other taxa or lineages, particularly those with smaller effective population sizes like humans, would be a fruitful avenue for future research.

Overall, our study provides a general and easy-to-use pipeline to identify lineage-specific de novo gene candidates. We also provide a systematic overview of the protein foldability of *Drosophilinae* lineage-specific de novo gene candidates. We propose that de novo genes undergo gradual sequence and functional adaptation without major protein structure changes in *Drosophila* lineage. With recent advances in de novo gene detection frameworks, such as those used in this work and in the work of others[33], it would be exciting to identify not only young but slightly older de novo genes in other lineages. This would provide an evolutionary framework for comparing de novo genes in multiple taxa. With more high-quality genome sequencing data, and transcription and translational data for more related species in the future, we will be able to identify de novo genes with higher confidence and uncover more about their emergence and how they were incorporated into the genomes and interactomes.

## Methods

### Identification of de novo gene candidates in *D. melanogaster*

To infer synteny, we used Progressive Cactus Genome Aligner[36] to align the genomes of 20 *Drosophilinae* species and 2 *Acalyptratae* species (Fig. 1a). To infer homology between different protein-coding genes, we also ran all-vs-all blastp using all the protein sequences of 20 *Acalyptratae* species along with another 8 *Arthropods* species (Fig. 1a). After obtaining the synteny and homology information, we used the following workflow (Fig. 1b) to identify possible candidates of de novo genes.

1.  As mentioned above, we used progressive Cactus aligner to align the 20 genomes of species in *Acalyptratae* lineages. The species

were assigned different branch numbers according to their separation from *D. melanogaster*, ranging from Br$_1$ to Br$_9$ (Fig. 1a). In this step, there were 13,798 *D. melanogaster* protein-coding genes aligned in Cactus whole genome alignments.

2. For each of annotated query protein-coding gene in one of the 20 *Acalyptratae* species, we used pairwise halLiftover to determine the syntenic region in another *Acalyptratae* species.

3. If the syntenic region is an annotated protein-coding gene, we assigned it to the ortholog of the query when the annotated gene has an E-value smaller than 0.05 to the query in the all-vs-all blastp analysis. Otherwise, if the syntenic region is unannotated, we used Genewise[39] and Spaln[40] to predict protein-coding potential from the unannotated syntenic region using the query gene as the reference. We assigned the syntenic hit as an unannotated ortholog if the coding potential was significantly beyond random simulations (see next section, *Random simulations of Geneise/ Spaln*, for details), otherwise we assigned it as a non-genic hit.

4. For each *D. melanogaster* protein-coding gene, we inferred their annotated/unannotated orthologs and non-genic hits based on the pairwise halLiftover in Step 2 and ortholog identification in Step 3. We were able to identify 1285 potential de novo gene candidates due to the presence of non-genic hits in their furthest aligned branches in Cactus alignments.

5. We assigned the query as a possible de novo gene candidate and inferred the origination branch to be Br$_i$ (i ∈ {0...9}) only when it has: (1) annotated or unannotated orthologs and homologous in branches Br$_i$ (Fig. 1a), (2) non-genic hits in the outgroups of Br$_i$, and (3) no homologs in the 8 distant *Arthropods* species as shown in Fig. 1a. To this preliminary step, we identified 686 potential de novo gene candidates, each with the inferred origination branch as the branch that has the most distant orthologs (Br$_i$ mentioned above and main text). These candidates were then subjected to further searches in all other species in UniprotKB and NCBI representative genomes (see below).

6. We then used jackhmmer[41] and blastp to search against UniProtKB sequence database to further filter out candidates that have homologs in species more distant than the inferred origination branch Br$_i$ as defined in step 5 and main text. To control for the possibility of false positives, which were quite frequent in iterative profile searches, we conducted three independent searches as follows:

   i. blastp search with E-value cutoff of 0.05.
   ii. Iterative jackhmmer with options "--incE 1e-5 -E 10".
   iii. Iterative jackhmmer with options "--incE 1e-5 -E 10". To control for false positives, after each iteration, we manually built hmm profile for next iteration by removing possible false positives with best 1 domain E-value larger than 1e-5[69]. For each de novo gene candidate, we stopped the search once the search converged or reached 5 iterations.

   To further control for possible false positives, for each de novo gene candidate, we required a reliable homolog to appear in at least two of the above searches with a E-value cutoff of 0.001. At this, we removed candidates that have reliable homologs in species that are more distant than their inferred origination branch obtained in step 5. The removed candidates, along with representative reliable homologs can be found in Supplementary Data 4.

7. As a final step, we used tblastn to search for possible unannotated homologs in species that are more distant to the inferred origination branches as defined in step 5 and main text. First, we used tblastn to search against NCBI representative genomes at E-value cutoff of 1.0. We then extracted the DNA sequences of the significant hits with the following command,

blastdbcmd -db ref_euk_rep_genomes -entry RefSeqID -range START-END -strand STRAND -out out.fasta -outfmt %f
where RefSeqID, STRAND, and START-END defined the locations of the significant hits. We extended the range (START-END) to match the size of the query *D. melanogaster* protein-coding genes. To further determine whether the tblastn hits were possible homologs, we used Genewise/Spaln to predict protein coding potential from the significant tblastn hits using the query *D. melanogaster* protein-coding gene as the reference. We manually examined the Gewise/Spaln predictions. A significant tblastn hit was considered as an unannotated homolog if it met the following criteria:

i. The predicted gene has a canonical start and a stop codon.
ii. The predicted gene has no frameshifts.
iii. The predicted gene has the same number of exons and introns as the query *D. melanogaster* gene or its orthologs in other *Drosophila* genomes.

At the tblastn filtering step, we removed candidates that have reliable unannotated homologs in the outgroup species of their inferred origination branches obtained in step 6. The removed candidates, along with representative reliable homologs can be found in Supplementary Data 4.

8. After the above filtering steps (step 6 and 7), we were able to identify 555 de novo gene candidates in *D. melanogaster* that potentially originated within *Drosophilinae* lineage. The full list of the 13968 annotated protein-coding gene in *D. melanogaster*, 13798 aligned in Cactus, 1285 potential candidates with non-genic hits, 686 preliminary candidates, and final 555 de novo gene candidates can be found in Supplementary Data 5.

For each de novo gene candidate identified here, we used the longest protein isoforms as the protein sequence for further analysis.

## Micro-synteny and orthoMCL analysis

We applied MCScanX[70] to perform pairwise synteny analysis with a relaxed micro-synteny option similar to Vakirlis et al.[33]. We plotted the chromosomal regions in *D. melanogaster* with protein-coding genes mapped to other genomes in the pairwise MCScanX analysis (Fig. S8). We further used orthoMCL[71] to obtain the ortholog groups between the species examined in our study (Fig. 1) with blastp E-value cutoff of 0.05 and percent of match cutoff of 0.5. We then compared the orthologs recovered by orthoMCL, Cactus aligner, and MCScanX with micro-synteny option (Table S4).

## Random simulations of Genewise/Spaln

We used random simulations to calculate the random expectations of Genewise/Spaln gene structure predictions. The random simulations were carried out as follows:

1. We simulated situations with different protein length, ranging from 15, 20, 25, 50, 75, 100, 200, 500 to 1000.

2. For each protein length (*N*), we ran 10,000 random simulations. For each simulation, we generated a random protein sequence according to the ratio of each amino acid in *D. melanogaster*, and a random DNA sequence with 3*(N + 300) base pairs according to the genome-wide GC content of *D. melanogaster* protein-coding DNA sequences. We performed Genewise or Spaln gene structure predictions using the proteins as references and the DNA sequences as targets. To assess the results of the predictions, we used the score reported by spliced alignment, as the score includes the penalty for introducing splicing sites. We calculated the mean (Mean) and standard deviation (SD) of the 10,000 random simulations.

3. We fitted Mean and SD as functions of protein length (N) respectively, using a two-phase decay function using Lmfit[72]

$$S = S0 + SpanFast* \exp(-KFast*N) + SpanSlow* \exp(-KSlow*N) \quad (1)$$

where

$$SpanFast = (S0 - S\min)*PercentFast*0.01 \quad (2)$$

$$SpanFast = (S0 - S\min)*(1\text{-}PercentFast)*0.01 \quad (3)$$

S0 and Smin were the expected maximum and minimum spliced alignment scores. We then can predict the random expectation scores for Genewise/Spaln gene structure prediction score for any protein references with length of N by the two-phase decay function (Fig. S10).

4. We computed the probability of the protein reference and DNA target being completely unrelated or non-homologous using one-tailed Gaussian distribution with the inferred mean and standard deviations of Genewise/Spaln prediction scores. We inferred the target DNA sequences being potentially genic and homologous to the reference protein at a *P*-value of $10^{-6}$.

## Bioinformatic analysis and evolutionary analysis

We used RepeatMasker[73] and Dfam[74] to search for DNA repeats, including transposable elements (TEs) and simple repeats. We used ORFfinder to detect all putative ORFs that are intergenic and longer than 75 nt (coded for 25 amino acids). We included nested ORFs to better understand possible residue composition of these intergenic putative ORFs. For protein property predictions, we used deepcnf[31] to predict per residue probability of helix, sheet, coil, and solvent accessibility, AUCPreD[75] to predict structural disorder, and PredMP[76] to predict transmembrane probability. These properties were further normalized by protein length. These structural property predictors have been shown to have high accuracy compared to other methods[30,31,76,77]. For example, in the critical assessment of protein intrinsic disorder prediction by Necci et al.[30], the authors found that AUCPreD, along with flDPnn[78], were consistently among the top five predictors[30]. To further rule out the bias from the structural disorder predictors, we further used flDPnn, language model-based predictor ADOPT[79], and AlphaFold derived predictor AlphaFold_disorder[80], to predict the structural disorder for de novo proteins. We used signalp-5.0[32] to determine the probability of a protein containing a signal peptide. Sequence evolutionary rate (ω), adaptation rate (ω_a), and nonadaptation rates (ω_na) of de novo gene candidates were obtained from a previous study[65].

## Gene expression patterns

We downloaded gene expression profiles from FlyAtlas2[81]. We converted FPKM to TPM by normalizing FPKM against the summation of all FPKMs as follows:

$$TPM_i = \frac{FPKM_i}{\sum FPKM_j} \times 10^6 \quad (4)$$

After TPM conversion, we only retained genes with expression levels larger than 0.1 TPM for further analysis. We treated male and female whole-body TPM as male and female expression levels. To describe male specificities of *D. melanogaster* genes, we first calculated

Z-score by:

$$zscore = \frac{TPM(\text{male expression}) - TPM(\text{female expression})}{\sqrt{sd^2(\text{male expression}) + sd^2(\text{female expression})}} \quad (5)$$

We then calculated a normalized Z-score by:

$$zscore(\text{norm}) = \frac{zscore - \min(zscore)}{\max(zscore) - \min(zscore)} \quad (6)$$

To characterize tissue specificities, we calculated tau values[82] based on the expression profiles of 27 different tissues.

## Structural prediction for de novo gene candidates

We combined AlphaFold2[19], RoseTTAFold[22], trRosetta[21] and ESMFold[20] to predict the foldability of the de novo gene candidates. Specifically, for AlphaFold2 predictions, if the de novo gene was present in AlphaFold Protein Structure Database (AlphaFold DB), we downloaded the predicted structure from the database, otherwise we ran AlphaFold v2.0.0 scripts (https://github.com/deepmind/alphafold/tree/v2.0.0) to predict the protein structure. For RoseTTAFold predictions, we used the script *run_e2e_ver.sh* to predict the protein structures. For ESMFold, we used the api on the ESM Metagenomic Atlas website (https://esmatlas.com/about#api) to fold the protein sequences. We used the per-residue confidence score (pLDDT) from AlphaFold2, RoseTTAFold, and ESMFold predictions to predict the foldability of the de novo genes. For trRosetta predictions, we followed similar procedures as DeepMSA[83] and iteratively searched through Uniclust30[84], UniRef90[85], metaclust[86] and tara[87] protein sequence databases at default E-value cutoff of 1e-3 to collect homologous or distantly related homologous sequences. Specifically, at the first stage (Uniclust30), we used hhblits[88] to search against Uniclust30. At each stage from stage 2 to stage 4 (UniRef90, metaclust, and tara), we first used jackhmmer[41] to search against the database and then used hhblits against a custom database built from the sequences generated by jackhmmer. After each stage of searching, we used MAFFT to construct a multiple sequence alignment (MSA) to ensure that the MSA is of high quality. With the final MSA as input, we generated 100 parallel structural models by trRosetta[21]. To characterize the foldability of the de novo genes and the convergence of the predictions, we selected an ensemble of 20 structural models with the lowest potential energies and calculated the average pairwise TM-score[89] by TM-align[90]. To better align the PLDDT metric in AlphaFold2, RoseTTAFold, and ESMFold, the final foldability or convergence values were multiplied by 100.

## Ancestral sequence reconstruction

For each de novo gene, we extracted its orthologs and unannotated putative orthologs (if there are any) from the above analysis (see section *Identification of* de novo *gene candidates in D. melanogaster* in Methods). We used MAFFT-LINSI[91] to align the orthologous protein sequences. An initial guide tree was built based on the phylogeny relationships between the species that had orthologs. We then used the AAML module in PAML 4.9[92] to reconstruct the most probable ancestral sequences based on the alignments and the initial guide tree. For each branch, we only considered amino acids that had more than 50% coverage among the species in this branch, as the reconstruction can be unreliable at sites with alignment gaps. We further used AlphaFold2 and ESMfold to predict the structures of the most probable ancestral sequences.

## Molecular dynamics simulations

Starting from the predicted structural models of the 19 potentially well-folded de novo gene candidates and their most probable ancestral

states, we set up the molecular dynamics simulations using the GROMACS-2019[93] and the Amber ff99SB-ILDN force field[94]. For each system, we placed the protein in a dodecahedral box, with a minimum distance between the solute and the box boundary of 1.2 nm. We filled the simulation box with TIP3P water molecules and additional $Na^+$ and $Cl^-$ ions to neutralize the system and reach the 0.15 M salt concentration. We further minimized the potential energy of the system by the steepest descent method followed by conjugate gradient method until the maximum force was smaller than $200\,kJ\,mol^{-1}\,nm^{-1}$. For each simulation system, we performed three independent 200 ns MD simulations. We clustered the conformations from the last 100 ns trajectories of all the three independent MD simulations into 5 clusters by the density peaks algorithm[95]. To obtain structural fold similarity during MD simulations, we used TMalign[90] to compute the pairwise TM-score of the centers of the 5 clusters. We also referred to the center of the 5 clusters as representative structural models and the center of the top cluster as the MD refined structural model during MD simulations.

## Clustering of testis-biased genes

For each testis-biased gene, we obtained the scaled expression levels at different germline cell development stages, including (1) Early spermatogonia, (2) Late spermatogonia, (3) Early spermatocytes, (4) Late spermatocytes, (5) Early spermatids, and (6) Late spermatids. We further applied principal components analysis (PCA) on the scaled expression dataset to reduce the dimensionality. We then applied a k-means clustering algorithm to cluster these genes using the first three PCs, which could explain 82.0%, 12.2%, and 3.2% of the variants, respectively. The testis-biased genes were finally clustered into four clusters by k-means clustering method (Fig. S6). The sum of the squared error of k-means clustering was also shown in Fig. S6.

## Reporting summary

Further information on research design is available in the Nature Portfolio Reporting Summary linked to this article.

## Data availability

Whole genome alignments and genome versions used in this study are available at Figshare https://doi.org/10.6084/m9.figshare.19989395, along with the raw alignments and spliced alignments of de novo gene candidates and their non-coding hits in outgroup species. Versions and accession codes of these genomes are also available in Supplementary Data 1. Source data is provided as a source data file. Source data are provided with this paper.

## Code availability

Code and scripts for this study are available on GitHub (https://github.com/LiZhaoLab/DrosophilaDenovoGene) as well as on Zenodo (https://doi.org/10.5281/zenodo.10233859).

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

## Acknowledgements

We thank members of the Zhao laboratory for helpful discussions, and Nicolas Svetec, Eli Woloshin, and Vivian Yan for critically reading the manuscript. We thank Jason Banfelder, Bala Jayaraman, and The Rockefeller University High Performance Computing (HPC) Center for their support in computation. We also thank Evan Witt, Nicolas Svetec, Sylvia Durkin, and Alice Gadau for their help with ATAC-seq data and analysis. This work was supported by National Institutes of Health (NIH) MIRA R35GM133780, the Robertson Foundation, an Allen Distinguished Investigator Award from Paul G. Allen Family Foundation, a Rita Allen Foundation Scholar Program, and a Vallee Scholar Program (VS-2020-35) to L.Z. J.P. was supported by a C. H. Li Memorial Scholar Fund Award at The Rockefeller University. The content of this study is solely the responsibility of the authors and does not necessarily represent the official views of the funders.

## Author contributions

J.P. and L.Z. conceived the study, designed the experiments, and formulated the analyses. J.P. conducted all analyses. J.P. and L.Z. authored the paper.

## Competing interests

The authors declare no competing interests.
