## [Peer Review File · Nature Communications]

REVIEWER COMMENTS

Reviewer #1 (Remarks to the Author):

Overall, this study makes a useful and interesting contribution to the field of de novo gene evolution. It introduces a sensible and transferrable framework for detecting orthologs of potential de novo genes in a focal species that relies on both BLAST and synteny-based approaches. One novelty is the use of whole-genome alignments. A key step in determining whether a gene truly has a de novo origin is being able to identify the non-coding sequence in outgroup species from which the gene originated, and the use of whole-genome alignments makes doing so more feasible in a good number of cases. Reviewers with more bioinformatic expertise should weigh in on the simulation-based method the authors propose for determining whether a syntenic region without an annotated protein-coding genes contains a likely ortholog.

After introducing this method, the rest of the study uses a variety of bioinformatic tools to measure different biochemical, gene expression and evolutionary attributes of the list of de novo genes. One notable advance is the use of AlphaFold2 and other modern algorithms to glean structural insights into the new list of de novo genes, though the findings (e.g., that the structures of the encoded proteins largely don't change after gene birth) to some extent confirm at scale what previous work had already hypothesized (as the authors note at lines 41, 371). A similar story exists for the other analyses: as prior efforts to identify de novo genes have also observed, the authors find that de novo genes have higher levels of intrinsic structural disorder, a frequent expression bias toward the testes (and ovary and brain), and higher rates of protein evolution. Whether the current findings are novel or confirmatory perhaps hinges on how much overlap there is between this manuscript's set of de novo gene candidates and the sets identified by previous work (e.g., Heames et al. 2020 JME for *Drosophila*), a point on which the authors could be invited to elaborate in revision. If the current work is analyzing a substantially non-overlapping and much higher confidence set of de novo genes (e.g., because their detection method is more discerning and/or they are working with better genome assemblies), then that increases the impact of the work. Even if the work represents an advance for *Drosophila*, though, some additional discussion is warranted re: how similar the present manuscript's biochemical/expression/evolutionary trends are to those predicted for other de novo genes in other taxa.

Major points:

(The paragraph above is point 1.)

2. The analysis pipeline could be described more clearly and accessibly. Please clarify when the blastp all-vs-all (line 451) comes into play, since the stepwise description beginning on line 455 frames the analysis as mostly synteny based. It seems like blastp is used only to confirm that an annotated gene found in the syntenic region of a non-mel species is a likely ortholog, but what if the ortholog is present in a different region in a non-mel species? How is that dealt with? Or, is that what is meant when the authors write "Annotated coding genes that do not meet this criterion might be involved in other evolutionary mechanisms... and are not considered in this work" (line 460-462)? (Does "annotated coding genes" mean the *Dmel* gene, or the gene in the syntenic region of a non-mel species?) More generally, how

does the pipeline deal with lineage-specific duplications of de novo genes, either in *Dmel* or in other lineages? Is the *Dmel* gene just excluded from future analysis (as suggested on line 98-99)?

3. There seems to be confusion regarding the evolutionary patterns of GC content. Line 142-143 sets up the hypothesis that “translational selection on codon usage plays an important role” in the evolution of de novo genes. Canonically, this statement would imply that GC content should increase over time for improved translational efficiency/accuracy. This expectation isn’t born out according to lines 188-190, which report higher GC content in genes with more recent origins. But lines 419-420 state that “compared to younger de novo genes, older de novo genes have higher GC content.” And then things reverse again on line 422, which states: “GC content decreases with the age of the gene.” Please clarify these points. And, if GC content really does decline with gene age, what might explain that pattern?

4. At line 158, the authors conclude from their bioinformatic analysis of protein properties that de novo genes might have specific molecular/cellular functions and cite Vakirlis, Acar et al. 2020 to support the point. (The same citation recurs in the discussion, line 430.) The Vakirlis paper is great, but it finds beneficial effects of some emerging de novo proteins (in yeast) only by artificially overexpressing them. Additional relevant work to the *Drosophila* analysis here would be actual demonstrations of cellular functions and fitness effects of de novo genes in *Drosophila*, as shown in Lange et al. 2021 *Nat Comms* and Rivard et al. 2021 *PLOS Genetics*.

5. The data show (Fig. 2G and line 191) that male specificity of de novo gene expression is positively correlated with gene age. A similar trend was observed by Palmieri et al. (2014, *eLife*), which showed that the youngest orphan genes (likely including many de novo) found only in *D. pseudoobscura* had lower levels of expression bias than orphan genes that were slightly older (present in a few species). That group interpreted the finding as suggesting that de novo genes with male-specific expression were more likely to be retained. Does the authors’ data support a similar conclusion?

6. On lines 323-325, the authors write that while “previous studies emphasized the importance of mid-to-late spermatogenesis in de novo gene origination,” the results here suggest that de novo genes might also be expressed in an important way in early spermatogenesis. Specifically, they find that the younger de novo genes in this study are enriched in the early germline stages (lines 320-321), while the older ones are expressed more commonly in mid-to-late stages. I am curious about how to reconcile these results with their own group’s previous findings in Witt et al. (2019, *eLife*). That study examined de novo genes found only in *D. melanogaster*, i.e., genes that should be younger than any of the genes described here. It concluded that these genes were enriched in mid-to-late spermatogenesis. The current study finds genes that are older than this (found in a few species, i.e. originated at L1, L2, etc.) are more likely to be expressed early, but even older genes are more likely to be expressed late. So, it seems the very youngest genes are enriched in mid/late spermatogenesis, the next youngest in early spermatogenesis, and the older genes again in mid/late. How are the authors thinking about this evolutionary pattern? Also, how do the authors interpret all of their transcript expression data in light of the general finding that translation in fly testes can occur quite a bit later than transcription through a variety of post-transcriptional control processes (work from Fuller, White-Cooper, etc. labs)?

7. Line 428-9: The authors propose “that, upon origination, de novo genes were involved in certain molecular functions with strong fitness effects.” A reasonable argument for these genes’ eventual fitness effects is their continued presence in the *D. melanogaster* genome. But I am not sure there is evidence to support the idea that all or even most de novo genes have strong fitness effects immediately at or

shortly after their birth. Many models suggest a gradual accumulation of essential functions and interactions (Carvunis and Bornberg papers). A high death rate of orphan genes has been demonstrated in other *Drosophila* (Palmieri et al. 2014 eLife), suggesting that many de novo genes are expendable quickly after emergence. And, in the context of testis-expressed de novo genes, the vast majority of those functionally tested for fertility effects appear to be expendable under at least some experimental conditions (Rivard et al. 2021 PLOS Genetics). In light of these data, please re-consider the hypothesis/wording here.

Minor points and wording issues:

8. It would be useful to include a supplemental table that lists all 13,798 *D. melanogaster* genes that went into the analysis, and whether they were determined to be non-de novo [within the set of species examined], de novo, or dropped from the analysis for some of the reasons listed in the text.

9. The organization in the first Results paragraph is confusing. The authors begin with 1285 candidates that were aligned to previously unannotated regions in distant lineages (line 89) and end with the 589 de novo genes that come out of the pipeline and are presented in File S1 (line 104). In between, though, there is discussion of 73 genes with no annotated orthologs, of which 26 are found to be likely de novo (lines 93-97). It seems like these 26 are a specific subset of the 589, so might the paragraph read more smoothly to describe them after explaining how the 589 were identified?

10. It would be useful to list (in Fig. 2 or in a table) the number of de novo genes that fall into each age category (L1-L9).

11. Line 252: Re-word this conclusion. The results presented show that most de novo proteins are not likely to be well folded (554/589, line 220), and most that are well-folded into complex structures (16/19, line 245) adopt familiar structural folds. Thus, the concluding sentence (line 252) that “de novo genes might be well-folded with novel structural folds” is based on 3 examples out of nearly 600 proteins analyzed, and thus seems overly broad.

12. Small editing points:

-Line 72-73: “...589 de novo protein-coding genes in *D. melanogaster* that were born at least ~67 million years” – should “at least” be “within the last”?

-Line 75: “characterize” could perhaps be “predict,” reflecting the computational nature of the analyses.

-Line 144: “de novo genes” should be “de novo proteins,” as protein attributes are subsequently described.

-Line 162 (Fig. 2 legend): “GC content[] of each amino acid[]” – should refer to codons instead, not amino acids, since nucleotide sequences have G/C content, not proteins.

-Line 275: reference should be to Fig. 4C, not 4D.

-Line 298 and following: please clarify what “unneglectable” means/implies. Important? Previously overlooked?

-In Fig. S1, there is no explanation for why there are multiple data points at each Li line.

-Fig. 2 presents the order of the panels in an inconsistent way: by row, the order is A-B-C, D-E, but then F-H and G-I. Would it make more sense to end F-G, H-I, and/or to rearrange the final few panels to put the correlations graph (currently panel G) as the last panel?

-The references section has some duplicate entries (e.g., Abrusan 2013, Almargo 2019, Baek 2021...).

Reviewer #2 (Remarks to the Author):

Origin and structural evolution of de novo genes in *Drosophila*

In this manuscript, Peng & Zhao describe a comprehensive survey of evidence for de novo genes in *Drosophila* species genomes and characterise various properties of these genes. Overall I find this to be a useful and informative study, but there are several areas that need attention.

The manuscript has several main components: detection of de novo genes across *Drosophila* species; examination of the structural properties of the inferred proteins in modern genomes and inferred ancestral sequences; examination of sequence, expression and evolutionary properties of the inferred de novo genes.

Major points:

The authors argue that their whole genome alignment method has advantages over synteny analysis, but the benefits are not clear to me. The authors correctly describe the limitations of the synteny approach (line 339-340) - that it breaks down when the genome is too rearranged or diverged - but I don't see how the whole genome alignment doesn't suffer from the same issue. If the genomes are rearranged, the alignment will also be patchy. Could the authors please clarify what happens for the genome alignment when the genomes are highly rearranged. How does the whole genome alignment approach solve these problems?

line 92-93: the authors state that their method results in the exclusion of species-specific candidate de novo genes. Why? How? This isn't obvious to me. This seems like an important point to clarify.

The authors analyse G+C content of the candidate de novo genes. They find that de novo genes have lower G+C content. However, statistically speaking there should be more numerous (though shorter) ORFs in G+C poor regions (McLysaght & Hurst, 2016), which might create an ascertainment bias. Have the authors considered this? Might there also be an influence of lower recombination rates in low G+C regions? Could this affect the alignments and the ability to detect these de novo genes?

The authors appear to jump to the conclusion that the GC difference between de novo and established genes relates to translational selection (line 142), but I think this is an over-interpretation of scant evidence. This could be explained by an ascertainment bias, so the idea of translational selection would need to be tested before it can be claimed.

The same issue comes up on line 188 onwards. I think that exploring the idea of translational selection is interesting, but would need additional testing. For example, is the classic relationship with gene

expression level found? Are the codons indeed preferred codons? If the authors are not thinking of translational selection manifesting as codon usage bias then what kind of translational selection are they invoking? How does the gene G+C content relate to that of the surrounding genomic DNA? I think this needs deeper inspection and greater justification and explanation.

The section on folds and well-folded proteins -- line 205 onwards (page 8 by my reckoning, though the pages are not numbered) lacks some important information in my opinion. "MD simulations" are not explained or introduced. One has to go to the methods to find that MD probably stands for 'molecular dynamics' and even still, the broad methodology/technique is not introduced. What does it really mean to say that the folds remain stable in these simulations? What is being tested here?

The authors rely on AlphaFold2 for their structure predictions. My understanding is that this is an ab initio (rather than homology modelling) method, so is theoretically capable of predicting structures for proteins with no available homologs. However, I am concerned regarding the interpretation of the differences (or not) between the predicted structure of the modern protein as compared to the inferred ancestral protein sequence (page 10 - line 254 onwards). I am guessing that in most cases there are very few substitutions. What power does AlphaFold2 have to possibly return different structures when the sequences are only slightly different? I believe that the method does use alignments in one of the steps, and these alignments will presumably be the same or highly similar for the ancestral sequences. Is it reasonable to expect that AlphaFold2 might be capable of inferring an alternative, or significantly altered structure for the inferred ancestral sequence? If not, then it is not reasonable to interpret the lack of major differences as reflecting anything of the true biological history of the genes and instead might be a limitation of the method? I am not confident that this is indeed a problem, so if the authors are aware that the method can indeed do what they hope it does, then I think it merits some mention. Either way, I think it would be important to detail the limitations of this approach.

I do not understand the relevance of the MD simulations in this section (line 269) and I don't know what the notation '200 ns' means with respect to the simulations. Perhaps with more information I would understand better and be convinced. At present, I simply find myself wondering whether or not the methods used actually have sufficient scope to infer alternative structures given the underlying sequence similarity. I would appreciate a better explanation of this approach to justify the interpretations.

The expression analysis (page 11; line 299 onwards) clusters the testis-biased genes into four clusters. However, in k-means clustering the number of clusters is decided in advance. How did the authors decide on making 4 clusters rather than any other number? What is the justification?

Line 309: The authors state that de novo genes in cluster 1 differ from de novo genes in other clusters. However, they don't state whether or not they are similar to other cluster 1 genes (ie non de novo genes with similar expression pattern). What is the basis for the interpretation? What is being tested here?

The interpretation regarding shifts in pattern of expression with de novo gene age needs greater justification. What might be the biological basis for this?

Minor points:

line 90: the term Li is introduced without explanation or expansion (it comes later, in the figure legend, but I think it needs to be in the text too). Furthermore, I think the choice of terminology here is a bit confusing. It seems to be referring to both a branch AND the clade defined by that branch. I think the term can only be one or the other. As it stands, I found it confusing. I also am unused to seeing the word 'lineage' used to refer to a clade, so that was also a bit confusing.

line 99: why was it necessary to remove cases where there have been translocations? Don't these get removed anyway in the later step that considers synteny?

line 142: I do not understand the interpretation that lower G+C somehow relates to "immature codons" and I don't understand what is meant by that phrase.

line 203: sequential -> sequence

line 240: I find the section heading to be a bit misleading as only 3/19 have potentially novel folds.

line 303: "We found that ..." - my understanding is that this isn't a finding, but is the result of the clustering. Rephrase.

The words unneglectable and non-neglectable are both used in various points in the manuscript. I think the authors perhaps mean non-negligible.

There are various points in the manuscript where there are small syntax errors. In all cases I was confident that I understood the intended meaning, so there was no impediment to understanding, but these should be fixed before final publication. I have not listed them all here, but I do provide a few examples:

line 32: 'born from scratch through previously non-genic DNA' should perhaps be 'born from scratch from previously non-genic DNA'

line 90-92 - this sentence doesn't make sense to me.

line 299 : bad syntax

line 348: taxonomy -> taxonomically

Response to reviewers' comments

Reviewer #1 (Remarks to the Author):

Overall, this study makes a useful and interesting contribution to the field of de novo gene evolution. It introduces a sensible and transferrable framework for detecting orthologs of potential de novo genes in a focal species that relies on both BLAST and synteny-based approaches. One novelty is the use of whole-genome alignments. A key step in determining whether a gene truly has a de novo origin is being able to identify the non-coding sequence in outgroup species from which the gene originated, and the use of whole-genome alignments makes doing so more feasible in a good number of cases. Reviewers with more bioinformatic expertise should weigh in on the simulation-based method the authors propose for determining whether a syntenic region without an annotated protein-coding genes contains a likely ortholog.

Response: Thank you for your positive evaluation, helpful suggestions, and comments. We have now revised the manuscript according to your comments. We hope the reviewer find our response and revision adequate.

After introducing this method, the rest of the study uses a variety of bioinformatic tools to measure different biochemical, gene expression and evolutionary attributes of the list of de novo genes. One notable advance is the use of AlphaFold2 and other modern algorithms to glean structural insights into the new list of de novo genes, though the findings (e.g., that the structures of the encoded proteins largely don't change after gene birth) to some extent confirm at scale what previous work had already hypothesized (as the authors note at lines 41, 371). A similar story exists for the other analyses: as prior efforts to identify de novo genes have also observed, the authors find that de novo genes have higher levels of intrinsic structural disorder, a frequent expression bias toward the testes (and ovary and brain), and higher rates of protein evolution.

Response:

As the reviewer mentioned, we cited the few very nice case studies about de novo protein structure and their potential evolutionary trajectories, such as Large et al. Nature communications. However, there are very few published papers on the protein structure evolution of de novo genes at 3D structure level, and nearly none at a systematical level. We hope the reviewer agrees with us that our understanding of the origin of de novo protein structures is still extremely limited. In this study, we performed a systematic study about not only very young de novo proteins but also older de novo proteins to provide a dynamic view of the structural evolution and maintenance.

Here we argue that one of our major points is on the evolution of de novo genes after their origination rather than just their special properties. For example, we performed evolutionary analysis and observed that younger de novo genes have higher adaptation rates, although younger de novo genes shared similar structural properties as older de novo genes. These suggested de novo genes might gradually shift their function and sequence compositions without significant structural changes due to structural constraints. In other words, our work suggested that sequence changes are much faster

than protein structure changes, highlighting the need to study whether protein structures provide constraints in sequence changes in the future.

We have now cited works that showed special properties of de novo genes. For example, Begun et al., 2007; Heames et al., 2020; Palmieri et al., 2014; Vakirlis, Acar, et al., 2020; Zhao et al., 2014. These works were focused on certain tissues, less divergent species groups, and other taxonomy (line 165-166 in revised manuscript).

Whether the current findings are novel or confirmatory perhaps hinges on how much overlap there is between this manuscript's set of de novo gene candidates and the sets identified by previous work (e.g., Heames et al. 2020 JME for *Drosophila*), a point on which the authors could be invited to elaborate in revision. If the current work is analyzing a substantially non-overlapping and much higher confidence set of de novo genes (e.g., because their detection method is more discerning and/or they are working with better genome assemblies), then that increases the impact of the work.

Response:

Thanks for bringing up this question, which gives us an opportunity to clarify the fundamental differences between our work and Heames et al. 2020. Heames et al studied de novo genes from orphan genes. In contrast, we aim to study the origin and evolution of de novo genes with different origination time frames, thus we identified many more de novo genes in *D. melanogaster* at more stringent criteria. Please see detailed differences below:

1. Our study is at a larger scale.

Heames et al. only studied orphan genes. For *D. melanogaster*, they detected 66 de novo genes out of 246 *D. melanogaster* specific orphan genes. Their results showed that some orphan genes might be de novo genes, which we agree.

In our study, we identified 555 *Drosophila* lineage specific de novo gene candidates in *D. melanogaster*, with different many origination branches (Br1, Br2, to Br9 as in Figure 1), which is a significant step forward to identify *Drosophila* lineage-specific de novo genes. The reason to identify all de novo genes with different birth ages is to be able to study changes in de novo sequence and structure properties with respect to age.

2. Our de novo gene detection method is different and more stringent.

Heames et al. used all-vs-all BLASTP and Phylostratigraphy to identify orphan genes. They then used TBLASTN and UCSC 27-way insect multiz alignment to check whether the orphan genes are likely to be *de novo*. It is fair to say that Heames et al. adopted approaches that could be termed as “genomic phylostratigraphy”, where the identification of de novo genes accounts solely on BLAST/DIAMOND/TBLASTN search. As reviewed by Van Oss et al, these “genomic phylostratigraphy” methods have some limitations. First, as pointed out by Van Oss et al. and McLysaght et al., the “results are dependent on BLAST search criteria. Because it is based on sequence similarity, it is often difficult for phylostratigraphy to determine whether a novel gene has emerged de novo or has diverged from an ancestral gene beyond recognition, for instance following a duplication event.” Thus, the search method could suffer from false positive and false negative issues. In our method, to be as concise as possible, we applied all-vs-all blastp,

hmmer, tblastn, and synteny inferred from cactus whole genome alignments. As we explained later to question 1 of reviewer 2's, cactus whole genome alignments is essential for the de novo gene detection in our manuscript. We wanted to make it clear here that the purpose here it not to criticize Heames et al. but to provide a perspective that the methodology is different.

3. Cactus and synteny analysis are more accurate to infer outgroup non-genic orthologous regions.

Heames et al. used significant tblastn results in unannotated regions as “non-coding homologous genomic regions”. While this might be largely true, we caution that regions in non-model species do not necessarily mean non-genic, since whether these unannotated regions are genic or not relies on the annotation quality of outgroup genomes. It can be difficult to depend on *tblastn* to find syntenic non-genic homologous sequences, which is a big challenge when identifying *de novo* genes. In our method, we used cactus alignments as outgroup non-genic ortholog DNA sequences to infer de novo gene origination, which takes both DNA sequence homology and synteny into consideration.

4. More than half of the de novo genes identified by Heames et al are in our list, despite different age assignment.

As indicated in 1, Heames et al. identified 66 *D. melanogaster* de novo genes from *D. melanogaster* orphan genes. We checked the overlap between the list between our de novo gene candidates and those from Heames et al. We downloaded the de novo genes from Heames et al source data at https://zivgitlab.uni-muenster.de/ag-ebb/de-novo/droso_de_novo/-/blob/master/clusters/genes.csv. We extracted the de novo genes from *D. melanogaster* and found that 49 of them are in our list, while the other 17 are not. This shows that we have general agreement and have some disagreement.

For example, FBgn0036311, has homologs in *Scaptodrosophila* species in our analysis, while it was identified to be de novo by Heames et al with age of ~37 Mya, significantly shorter than the separation between *D. melanogaster* and *Scaptodrosophila* species, which is around 65 Mya according to timetree.org. Another example was FBgn0266456, which was identified to be de novo by Heames et al. with an age of ~37 Mya, while we observed reliable homologs in *Lucilia cuprina* by *jackhammer* iterative search against UniProtKB database with evaluate 2.7e-32. The homolog protein in *Lucilia cuprina* is A0A0L0CJX0. Again, the noted disparity is not meant to discredit their work, but rather to emphasize that it remains technically challenging to identify 'older' *de novo* gene rigorously and robustly. This was also one of our motivations for using cutting-edge methods to tackle this important yet difficult problem.

Based on the above, our study was significantly different from Heames et al. We study the origination and evolution of *de novo* genes. We identified *de novo* genes in *D. melanogaster* with different origination ages at a larger scale since we are not focusing on orphan genes. In addition, we have more stringent criteria, which take both homology and synteny into account. We identified more *de novo* genes in the analysis because our methods can better infer non-genic outgroup homologs. Together our results and Heames et al. highlighted the need to visit important questions using different methodologies and the power of new methods (such as progressive cactus aligner) in fundamental biology.

Even if the work represents an advance for *Drosophila*, though, some additional discussion is warranted re: how similar the present manuscript's biochemical/expression/evolutionary trends are to those predicted for other *de novo* genes in other taxa.

Response

As we explained above and in the manuscript, we firmly believe that our work represents an important advance for *Drosophila* *de novo* gene research. For example, as the reviewer noted, we cited Bungard et al. 2017 and Lange et al. 2021 extensively in our manuscript. However, these work – carried out by careful scientists – call their *de novo* gene of focus “putative *de novo* genes”. This, as a side note, highlighted the difficulty in obtaining solid evidence for older *de novo* genes, because of the general difficulties in searching for homologous non-genic sequences in the outgroup species.

Thanks for your suggestion on the comparisons. There is little knowledge about the biochemical trend in *de novo* genes in multiple taxa. For the rest, we have added discussions. We added some discussions in the revised manuscript (line 462-489 and line 493-497 in revised manuscript).

“Our study presents a comprehensive analysis of *de novo* genes of various ages, offering a systematic overview on their origination and evolution. While previous investigations in different lineages have employed diverse methodologies (Begun et al., 2007; Carvunis et al., 2012; Heames et al., 2020; Knowles & McLysaght, 2009; Majic & Payne, 2020; Moyers & Zhang, 2016; Ruiz-Orera et al., 2015; Vakirlis et al., 2018; Witt et al., 2019b; Wu et al., 2011; Zhao et al., 2014; Zheng & Zhao, 2022), several characteristics of *de novo* genes appear to be consistent across multiple lineages. Notably, *de novo* genes tend to be relatively short in length and exhibit a strong enrichment in testis, displaying biased functions associated with this reproductive organ. Additionally, in mammals, the expression of *de novo* genes in the brain is relatively common (An et al., 2023; Ruiz-Orera et al., 2015; Wu et al., 2011). This is also observed in older *de novo* genes in *Drosophila*, although to a much less degree. Intriguingly, our research highlights the potential for the immune system to serve as another hotspot for *de novo* gene origination. However, further investigations are required to determine if this pattern holds true in other taxa or lineages. Furthermore, our investigation reveals that *de novo* genes predominantly exhibit a disordered nature, and this characteristic remains stable over the time frame examined. These findings align with the study by Schmitz et al. (Schmitz et al., 2018). In contrast, there are some inconsistencies in previous studies. For example, Carvunis et al. observed in yeast that protein structural disorder increased with gene age (Carvunis et al., 2012), and Wilson et al. observed in yeast and mouse that protein structural disorder decreased with gene age. Investigating whether this discrepancy arises from methodological differences or possesses biological relevance warrants further exploration. Our work is the first study to reveal structural conservation for well-folded protein-coding genes, whether this is a conserved pattern awaits future studies in other taxa or lineages. Another major point of contention revolves around the origin of *de novo* genes: whether they arise through neutral processes or are driven by strong selection. Our study on *D. melanogaster*, a species that has a large effective population size, demonstrates that both adaptive and nonadaptive changes play pivotal roles in the slightly accelerated evolution of *de novo*

genes after their birth. Previous studies that have distinguished adaptive and nonadaptive rates in de novo gene evolution are scarce. Exploring the applicability of our findings to other taxa or lineages, particularly those with smaller effective population sizes like humans, would be a fruitful avenue for future research.

... in *Drosophila* lineage. in *Drosophila* lineage. With recent advances in de novo gene detection frameworks, such as those used in this work and in the work of others, e.g., Vikirilis et al., 2020 (Vakirlis, Carvunis, et al., 2020), it would be exciting to identify not only young but slightly older de novo genes in other lineages. This would provide an evolutionary framework for comparing de novo genes in multiple taxa.”

Major points:

(The paragraph above is point 1.)

2. The analysis pipeline could be described more clearly and accessibly. Please clarify when the blastp all-vs-all (line 451) comes into play, since the stepwise description beginning on line 455 frames the analysis as mostly synteny based. It seems like blastp is used only to confirm that an annotated gene found in the syntenic region of a non-mel species is a likely ortholog,

Response

Thanks for bringing up this question. We apologize for the confusions. All-vs-all blastp is used in the following steps: 1) as the reviewer pointed out, we used *blastp* to determine if syntenic genes from all species studied are orthologs. 2) blastp also help identify homologous genes for multi-copy gene and gene family; these genes are not likely to be de novo genes but nonetheless an important part of the analysis. 3) in the case of single-copy genes that have translocation events, blastp can also help identify homologous or orthologous genes that do not share synteny. This is an important step to complement synteny-based analysis. By this iterative pipeline, we could reveal as much orthology information as possible.

To make it less confusing, we rephrased the pipeline section *Identification of de novo gene candidates in D. melanogaster in Material and Methods* as following (line 503-582 in revised manuscript):

To infer homology between different protein-coding genes, we also run all-vs-all *blastp* using all the protein sequences of 20 *Acalypratae* species along with another 8 *Arthropods* species (Figure 1A). After obtaining the synteny and homology information, we used the following workflow (Figure 1B) to identify possible candidates of *de novo* genes.

1. As mentioned above, we used progressive cactus aligner to align the 20 genomes of species in *Acalypratae* lineages. The species were assigned different branch numbers according to their separation to *D. melanogaster*, ranging from Br_1 to Br_9 (Figure 1A). In this step, there were 13798 *D. melanogaster* protein-coding genes aligned in cactus whole genome alignments.
2. For each of annotated query protein-coding gene in one of the 20 *Acalypratae* species, we used pairwise *halLiftover* to determine the syntenic region in another *Acalypratae* species.

3. If the syntenic region is an annotated protein-coding gene, we assign it to the ortholog of the query when the annotated gene has an E-value smaller than 0.05 to the query in the all-vs-all *blastp* analysis. Otherwise if the syntenic region is unannotated, we use *Genewise* (Birney et al., 2004) and *Spaln* (Iwata & Gotoh, 2012) to predict protein-coding potential from the unannotated syntenic region using the query gene as reference. We assign the syntenic hit as an unannotated ortholog if the coding potential was significantly beyond random simulations (see next section, *Random simulations of Geneise/Spaln*, for details), otherwise we assign it as a non-genic hit.
4. For each *D. melanogaster* protein coding gene, we inferred their annotated/unannotated orthologs and non-genic hits based on the pairwise *halLiftover* in Step 2 and ortholog identification in Step 3. We were able to identify 1285 potential *de novo* gene candidates due to the presence of non-genic hits in their furthest aligned branches in cactus alignments.
5. We assigned the query as a possible *de novo* gene candidate and inferred the origination branch to be Br_i ($i \in \{0 \dots 9\}$) only when it has: (1) annotated or unannotated orthologs and homologous in branches Br_i (Fig. 1A), (2) non-genic hits in the outgroups of Br_i , and (3) no homologs in the 8 distant *Arthropods* species as shown in Fig. 1A. To this preliminary step, we identified 686 potential *de novo* gene candidates, each with the inferred origination branch as the branch that has the most distant orthologs (Br_i mentioned above and main text). These candidates then are subjected for further searches in all other species in UniprotKB and NCBI representative genomes (see below).
6. We then used *jackhmmmer* (S. R. Eddy, 2011) and *blastp* to search against UniProtKB sequence database to further filter out candidates that have homologs in species more distant than the inferred origination branch Br_i as defined in step 5 and main text. To control for the possibility of false positives, which were quite frequent in iterative profile searches, we conducted three independent searches as follows:
 - i. *blastp* search with E-value cutoff of 0.05.
 - ii. Iterative *jackhmmmer* with options "--incE 1e-5 -E 10".
 - iii. Iterative *jackhmmmer* with options "--incE 1e-5 -E 10". To control for false positives, after each iteration, we manually built hmm profile for next iteration by removing possible false positives with best 1 domain E-value larger than 1e-5 (S. Eddy, 1992). For each *de novo* gene candidate, we stopped the search once the search converged or reached 5 iterations.

To further control for possible false positives, for each *de novo* gene candidate, we required a reliable homolog to appear in at least two of the above searches with a E-value cutoff of 0.001. At this, we removed candidates that have reliable homologs in species that are more distant than their inferred origination branch obtained in step 5. The removed candidates, along with representative reliable homologs can be found in *supplementary File S4: reliable blastp jackhmmmer and tblastn homologs*.

7. As a final step, we used *tblastn* to search for possible unannotated homologs in species that are more distant to the inferred origination branches as defined in step 5 and main text. First, we used *tblastn* to search against NCBI representative genomes at E-value cutoff of 1.0. We then extracted the DNA sequences of the significant hits with the following command,

```
blastdbcmd -db ref_euk_rep_genomes -entry RefSeqID -range START-END -strand STRAND -out out.fasta -outfmt %f
```

where *RefSeqID*, *STRAND*, and *START–END* defined the locations of the significant hits. We extended the range (*START–END*) to match the size of the query *D. melanogaster* protein-coding genes. To further determine whether the *tblastn* hits were possible homologs, we used *Genewise/Spaln* to predict protein coding potential from the significant *tblastn* hits using the query *D. melanogaster* protein-coding gene as reference. We manually examined the *Genewise/Spaln* predictions. A significant *tblastn* hit was considered as an unannotated homolog if it met the following criteria:

- i. The predicted gene has canonical start and stop codons.
- ii. The predicted gene has no frameshifts.
- iii. The predicted gene has the same number of exons and introns as the query *D. melanogaster* gene or its orthologs in other *Drosophila* genomes.

At the *tblastn* filtering step, we removed candidates that have reliable unannotated homologs in the outgroup species of their inferred origination branches obtained in step 6. The removed candidates, along with representative reliable homologs can be found in *supplementary File S4: reliable blastp jackhammer and tblastn homologs*.

8. After the above filtering steps (step 6 and 7), we were able to identify 555 *de novo* gene candidates in *D. melanogaster* that are potentially originated within *Drosophilinae* lineage. The full list of the 13968 annotated protein-coding gene in *D. melanogaster*, 13798 aligned in cactus, 1285 potential candidates with non-genic hits, 686 preliminary candidates, and final 555 *de novo* gene candidates can be found in *supplementary File S6: list of D. melanogaster protein-coding genes in the de novo gene identification workflow*.

but what if the ortholog is present in a different region in a non-mel species? How is that dealt with?

Response:

As the reviewer pointed out, if the ortholog is present in a different region in a non-mel species. We excluded that gene in further analysis since this could indicate translocation events or other mechanisms. As mentioned above, we have rephrased our identification pipeline (such as using *blastp*) to make the identification process clearer (line 503-582 in revised manuscript).

Or, is that what is meant when the authors write “Annotated coding genes that do not meet this criterion might be involved in other evolutionary mechanisms... and are not considered in this work” (line 460-462)?

Response:

The reviewer is right that these genes are not considered in this work. We have rephrased our pipeline section *Identification of de novo gene candidates in D. melanogaster* in *Material and Methods* as above.

(Does “annotated coding genes” mean the *Dmel* gene, or the gene in the syntenic region of a non-mel species?)

Response:

Sorry for the confusion. Here, the annotated coding genes are all annotated protein coding genes in all the species used in progressive cactus alignments. As we mentioned above, we have rephrased our pipeline section to make it less confusing in *Material and Methods*, section *Identification of de novo gene candidates in D. melanogaster*.

More generally, how does the pipeline deal with lineage-specific duplications of de novo genes, either in Dmel or in other lineages?

Response:

Thanks for bringing up this topic. Our de novo gene detection pipeline relies on progressive cactus whole genome aligner to identify lineage-specific duplications. Technically, progressive cactus builds ancestral genomes from closely related genomes that share the same internal node. The progressive cactus aligner uses two algorithms to resolve paralogy, namely “*single-copy outgroup filtering*” and “*best-hit filtering*”. The first *single-copy outgroup filtering* algorithm identifies all possible paralogs while the second “*best-hit filtering*” algorithm can remove ancient duplications. The program would then align lineage-specific duplications to the same ortholog region in ancestral genomes.

Figure adapted from Armstrong et al. 2020. In this figure, different colors represent bases in different species and edges represented pairwise alignment relationships. The thickness of the edges represents the alignment scores, where higher thickness suggests higher alignment scores. In the case of lineage specific duplication (green dots of copy B), cactus could resolve the correct ortholog and paralog information.

In our analysis, we found several de novo genes were involved in lineage/branch specific duplication events. To assess if our progressive cactus pipeline was able to preserve orthology and paralogy information, we did an orthoMCL analysis from the all-vs-all blastp results. In many cases, cactus results were consistent with orthoMCL.

For example, the two genes FBgn0051909|FBgn0264344 were identified as de novo genes candidates originated in Branch 1. They were paralogs as annotated in Flybase. In orthoMCL results, they were identified to be Dmel specific duplications, with their Dsim and Dsec orthologs to be GD23456, and LOC6611512, respectively. In our study, they were also identified as Dmel specific duplications. Their orthologs in Dsim and Dsec were also GD23456 in Dsim, and LOC6611512 in Dsec. In addition, progressive cactus was able to identify orthologs non-genic regions in species from Branch 2 (Dyak and Dere), all the way to Branch 9 (Sleb).

Another example is the following five de novo gene candidates originated in Branch 1 in our study: FBgn0053664, FBgn0053665, FBgn0053666, FBgn0053667, FBgn0053668, and FBgn0053669. The five genes were identified as paralogs in both orthoMCL results and our progressive cactus pipeline, with the same ortholog, GD28268, identified in Dsim.

Other examples include FBgn0029694|FBgn0037910, FBgn0085361|FBgn0260871, FBgn0051909|FBgn0264344, FBgn0264989|FBgn0264990, FBgn0265625|FBgn0265626|FBgn0265627, FBgn0261060|FBgn0262363, FBgn0261580|FBgn0261581, and FBgn0086915|FBgn0267411|FBgn0267412|FBgn0267413|FBgn0267417|FBgn0267490|FBgn0267491, etc.

In some other cases, progressive cactus was able to identify duplicates that were not identified in orthoMCL. For example, FBgn0029589|FBgn0029590, FBgn0051797|FBgn0051921|FBgn0264086, FBgn0259963|FBgn0265349, FBgn0032590|FBgn0051815, FBgn0033165|FBgn0033167, and FBgn0264746|FBgn0264747|FBgn0264748. In these examples, the paralogs were all similar to each other with e-values smaller than $1e-8$.

Since our main topic in current study is on the origin and structural evolution of de novo genes, how the de novo genes duplicate after origination would be beyond the scope of our current manuscript. We thank the reviewer for bringing up this issue and we agree that lineage specific duplication of de novo gene is a super exciting topic to study in the future.

Is the Dmel gene just excluded from future analysis (as suggested on line 98-99)?

Response:

The reviewer is right that these genes are not considered in future analysis since this could indicate translocation events or other mechanisms. To make the pipeline clearer, we have rephrased our pipeline section *Identification of de novo gene candidates in D. melanogaster* in *Material and Methods* as above.

3. There seems to be confusion regarding the evolutionary patterns of GC content. Line 142-143 sets up the hypothesis that “translational selection on codon usage plays an important role” in the evolution of de novo genes. Canonically, this statement would imply that GC content should increase over time for improved translational efficiency/accuracy. This expectation isn’t born out according to lines 188-190, which report higher GC content in genes with more recent origins. But lines 419-420 state that “compared to younger de novo genes, older de novo genes have higher GC content.” And then things reverse again on line 422, which states: “GC content decreases with the age of the gene.” Please clarify these points. And, if GC content really does decline with gene age, what might explain that pattern?

Response

Thanks for correcting the typo. We observed that GC content increases with gene age, we have revised Figure 2 and main text accordingly.

4. At line 158, the authors conclude from their bioinformatic analysis of protein properties that *de novo* genes might have specific molecular/cellular functions and cite Vakirlis, Acar et al. 2020 to support the point. (The same citation recurs in the discussion, line 430.) The Vakirlis paper is great, but it finds beneficial effects of some emerging *de novo* proteins (in yeast) only by artificially overexpressing them. Additional relevant work to the *Drosophila* analysis here would be actual demonstrations of cellular functions and fitness effects of *de novo* genes in *Drosophila*, as shown in Lange et al. 2021 Nat Comms and Rivard et al. 2021 PLOS Genetics.

Response

Thanks for the valuable suggestion. We have now added Lange et al. 2021 Nat Comms, Rivard et al. 2021 PLOS Genetics here (line 175 and line 450 in revised manuscript).

5. The data show (Fig. 2G and line 191) that male specificity of *de novo* gene expression is positively correlated with gene age. A similar trend was observed by Palmieri et al. (2014, eLife), which showed that the youngest orphan genes (likely including many *de novo*) found only in *D. pseudoobscura* had lower levels of expression bias than orphan genes that were slightly older (present in a few species). That group interpreted the finding as suggesting that *de novo* genes with male-specific expression were more likely to be retained. Does the authors' data support a similar conclusion?

Response

We observed *de novo* genes are more likely to be male-biased, which is in line with the comparison of orphan genes and older genes by Palmieri et al. We have added the reference in the revised manuscript accordingly (line 165 in the revised manuscript). However, for *Drosophila de novo* genes, we observed that male-specificity decreased with gene age significantly, while male-expression levels did not have significant correlations.

6. On lines 323-325, the authors write that while “previous studies emphasized the importance of mid-to-late spermatogenesis in *de novo* gene origination,” the results here suggest that *de novo* genes might also be expressed in an important way in early spermatogenesis. Specifically, they find that the younger *de novo* genes in this study are enriched in the early germline stages (lines 320-321), while the older ones are expressed more commonly in mid-to-late stages. I am curious about how to reconcile these results with their own group's previous findings in Witt et al. (2019, eLife). That study examined *de novo* genes found only in *D. melanogaster*, i.e., genes that should be younger than any of the genes described here. It concluded that these genes were enriched in mid-to-late spermatogenesis. The current study finds genes that are older than this (found in a few species, i.e. originated at L1, L2, etc.) are more likely to be expressed early, but even older genes are more likely to be expressed late. So, it seems the very youngest genes are enriched in mid/late spermatogenesis, the next youngest in early spermatogenesis, and the older genes again in mid/late. How are the authors thinking about this evolutionary pattern? Also, how do the authors interpret all of their transcript expression data in light of the general finding that translation in fly testes can occur quite a bit later than transcription through a variety of post-transcriptional control processes (work from Fuller, White-Cooper, etc. labs)?

Response

Our results align with Witt et al. 2019, which found that most *de novo* genes are enriched in late spermatocytes and spermatids. Spermatocytes and spermatids have been extensively studied in the literature, not only in our work but also in studies from other labs, such as Fuller, White-Cooper, and others. In this manuscript, we were particularly interested in why germline stem cells and early spermatogonia also have a substantial number of *de novo* genes. This pattern was indeed observed in Witt et al. 2019. Even though germline stem cells and early spermatogonia only constitute a small subset of cells in the testis, Witt et al. 2019 indicated that approximately 15% young *de novo* genes are enriched at this stage. Consequently, in this paper, we wanted to examine whether genes expressed in early cells differed from those in other cell types at the structural and evolution level.

Upon reconsidering the reviewer's feedback and that of another reviewer, we realized that our exclusive focus on this stage may be unjustified as the data points were not related to age analysis independent. This could potentially give readers the wrong impression. In fact, all *de novo* genes are enriched in later stages. Therefore, we decided to retain our analysis of the evolutionary and structural properties of Cluster 1, while omitting the age-related analysis.

The work by Raz et al. 2023 (doi.org/10.7554/eLife.82201) is extremely intriguing, and we have been actively considering what their discoveries imply within the context of gene evolution. In the future, we are interested in systematically comparing the same genes using both single-cell and single-nuclei data. We look forward to the possibility of single-cell transcriptome data becoming available at some point.

7. Line 428-9: The authors propose “that, upon origination, *de novo* genes were involved in certain molecular functions with strong fitness effects.” A reasonable argument for these genes' eventual fitness effects is their continued presence in the *D. melanogaster* genome. But I am not sure there is evidence to support the idea that all or even most *de novo* genes have strong fitness effects immediately at or shortly after their birth. Many models suggest a gradual accumulation of essential functions and interactions (Carvunis and Bornberg papers). A high death rate of orphan genes has been demonstrated in other *Drosophila* (Palmieri et al. 2014 eLife), suggesting that many *de novo* genes are expendable quickly after emergence. And, in the context of testis-expressed *de novo* genes, the vast majority of those functionally tested for fertility effects appear to be expendable under at least some experimental conditions (Rivard et al. 2021 PLOS Genetics). In light of these data, please re-consider the hypothesis/wording here.

Response

Thanks for bringing this up. We agree that *de novo* genes should have fast turn-over rate. Similar trends have also been reported in a recent paper from our group on unannotated ORFs in *Drosophila*. We have revised our hypothesis by replacing “strong” with “certain”. We have also added a discussion (line 447-448 in revised manuscript) to reflect the fact that some *de novo* genes might be expendable quickly after emergence. “In cases where *de novo* genes lost their functions, they might be depleted shortly after origination (Palmieri et al., 2014; Zheng & Zhao, 2022)”.

Minor points and wording issues:

8. It would be useful to include a supplemental table that lists all 13,798 *D. melanogaster* genes that went into the analysis, and whether they were determined to be non-de novo [within the set of species examined], de novo, or dropped from the analysis for some of the reasons listed in the text.

Response

Thanks for the valuable suggestion. We have added a supplementary table in supplementary_File_S5_summary_number_annotated.csv (line 578 in revised manuscript) to indicate number of genes that went into each step of our analysis.

9. The organization in the first Results paragraph is confusing. The authors begin with 1285 candidates that were aligned to previously unannotated regions in distant lineages (line 89) and end with the 589 de novo genes that come out of the pipeline and are presented in File S1 (line 104). In between, though, there is discussion of 73 genes with no annotated orthologs, of which 26 are found to be likely de novo (lines 93-97). It seems like these 26 are a specific subset of the 589, so might the paragraph read more smoothly to describe them after explaining how the 589 were identified?

Response

Thanks for bringing this up. Following the reviewer's suggestion, we have reorganized this paragraph. In the revised paragraph, we moved the description of the unannotated orthologs after explaining how the de novo genes were identified. Since we have added the *tblastn* filtering steps to filter out candidates with distant putative unannotated homologs, the number of de novo genes identified now is 555.

10. It would be useful to list (in Fig. 2 or in a table) the number of de novo genes that fall into each age category (L1-L9).

Response

Thanks for the valuable suggestion. We have now added a supplementary figure (Figure S2) in the revised manuscript. The supplementary figure is attached below. Corresponding descriptions were added in line 111-112 in the revised manuscript.

Figure S2. Number of *de novo* gene candidates identified in each branch. The number generally correlated with the divergence time between *D. melanogaster* and each branch.

11. Line 252: Re-word this conclusion. The results presented show that most *de novo* proteins are not likely to be well folded (554/589, line 220), and most that are well-folded into complex structures (16/19, line 245) adopt familiar structural folds. Thus, the concluding sentence (line 252) that “*de novo* genes might be well-folded with novel structural folds” is based on 3 examples out of nearly 600 proteins analyzed, and thus seems overly broad.

Response

It’s a great point. We have rephrased the section to reflect the main finding that most of the potentially well-folded adopt existing structural folds. We rephrased the title as “Most potentially well folded *de novo* genes adopts existing protein structure folds” (line 261 in revised manuscript). We also rephrased our concluding sentence as “Overall, our results indicated that well-folded *de novo* genes are likely to adopt existing protein structure folds” (line 272-273 in revised manuscript).

12. Small editing points:

-Line 72-73: “...589 *de novo* protein-coding genes in *D. melanogaster* that were born at least ~67 million years” – should “at least” be “within the last”?

Response

We have revised the wording accordingly (line 79-80 in revised manuscript). Thank you.

-Line 75: “characterize” could perhaps be “predict,” reflecting the computational nature of the analyses.

Response

We have revised the wording accordingly (line 82 in revised manuscript). Thank you.

-Line 144: “de novo genes” should be “de novo proteins,” as protein attributes are subsequently described.

Response

We have revised the wording accordingly (line 159 to “protein products of *de novo* genes” and line 404 to “*de novo* proteins” in revised manuscript). Thank you.

-Line 162 (Fig. 2 legend): “GC content[] of each amino acid[]” – should refer to codons instead, not amino acids, since nucleotide sequences have G/C content, not proteins.

Response

Thanks for the valuable suggestions, we have revised the wording accordingly. Specifically, this sentence was rephrased to “GC contents of the codons utilized by each of the amino acids in *de novo* gene candidates were significantly lower than those of the amino acids in other annotated protein-coding genes” (line 179-181 in revised manuscript).

-Line 275: reference should be to Fig. 4C, not 4D.

Response

Thanks, we have revised Fig. 4D to Fig. 4C.

-Line 298 and following: please clarify what “unneglectable” means/implies. Important? Previously overlooked?

Response

Thanks for the valuable suggestions, we have rephrased “unneglectable” to “non-negligible” here (line 346 in revised manuscript) and in the abstract to “potentially important but less emphasized” (line 25 in revised manuscript).

-In Fig. S1, there is no explanation for why there are multiple data points at each Li line.

Response

Thanks for bringing up this issue. As suggested by other reviewers, we have renamed lineage Li to branch Bri to better reflect we are studying species in the *Drosophila* lineage. The reason why there are multiple data points at each branch is that there are multiple species/genomes in each branch. These data points suggested the overall alignment coverage and divergence between *D. melanogaster* and each species in each branch.

-Fig. 2 presents the order of the panels in an inconsistent way: by row, the order is A-B-C, D-E, but then F-H and G-I. Would it make more sense to end F-G, H-I, and/or to rearrange the final few panels to put the correlations graph (currently panel G) as the last panel?

Response

We have swapped Figure 2G and Figure 2I in the revised manuscript.

-The references section has some duplicate entries (e.g., Abrusan 2013, Almargo 2019, Baek 2021...).

Response

We have now removed duplicate entries. Thank you.

Reviewer #2 (Remarks to the Author):

Origin and structural evolution of de novo genes in *Drosophila*

In this manuscript, Peng & Zhao describe a comprehensive survey of evidence for de novo genes in *Drosophila* species genomes and characterise various properties of these genes. Overall I find this to be a useful and informative study, but there are several areas that need attention.

The manuscript has several main components: detection of de novo genes across *Drosophila* species; examination of the structural properties of the inferred proteins in modern genomes and inferred ancestral sequences; examination of sequence, expression and evolutionary properties of the inferred de novo genes.

Response: Thank you for your evaluation and constructive comments. We have incorporated additional analysis, including the use of the ESMFold language model, to reevaluate results related to AlphaFold. Additionally, we have conducted a more in-depth analysis of GC content and thoroughly revised a large section of the methods. We appreciate the reviewer's thoughtful comments and careful reading of the manuscript.

Major points:

The authors argue that their whole genome alignment method has advantages over synteny analysis, but the benefits are not clear to me. The authors correctly describe the limitations of the synteny approach (line 339-340) - that it breaks down when the genome is too rearranged or diverged - but I don't see how the whole genome alignment doesn't suffer from the same issue. If the genomes are rearranged, the alignment will also be patchy. Could the authors please clarify what happens for the genome alignment when the genomes are highly rearranged. How does the whole genome alignment approach solve these problems?

Response:

Thanks for bringing up this point.

Synteny based methods, such as MCScanX (Wang et al. 2012), build synteny mapping using protein coding genes as anchors. In this case, it would require at least three consecutive protein coding genes to define synteny or micro-synteny maps. While in progressive cactus aligner (Armstrong et al. 2020) starts with aligning two genomes by sensitive pairwise local alignments from LASTZ (Harris et al. 2007). These pairwise alignments were further filtered and refined to create a more complete alignment (Armstrong et al. 2020). Note that by creating a set of pairwise local alignments, progressive cactus aligner creates many more tiny "anchors" compared to conventional synteny based method. These tiny "anchors" could be in conserved protein-coding genes, non-coding genes or even intergenic regions. The anchors were further scored and filtered to best represent the whole genome alignments. This could potentially result in more precise micro-synteny maps and these micro-synteny maps could potentially help in the case where genomes are highly rearranged. It was also discussed elsewhere

that cactus can generally capture homologies well in the presence of rearrangements as assessed by Alignathon, a competitive assessment of whole-genome alignment methods (Earl et al. 2014, 10.1101/gr.174920.114; Armstrong et al. 2018, 10.1146/annurev-animal-020518-115005).

To further test whether our progressive cactus pipeline would perform better than synteny based methods, we compared the orthologs that can be identified and aligned by cactus, since the identified or aligned orthologs could serve as evidence of micro-synteny maps. First, we ran an orthoMCL analysis to search for all possible ortholog pairs of protein-coding genes in the species we studied, with default settings (E-value cutoff 1e-3 and percent of match cutoff 50%). For the synteny based method, we applied MCScanX using micro-synteny option, with blastp E-value cutoff set to 0.05, and the number of genes required to all synteny set to 3. We term this option micro-synteny here since similar micro-synteny analysis was performed by Vakirlis et al. 2020 to study the divergence of orphan genes.

By comparing the identified orthologs, we found that cactus can have comparable performance with orthoMCL in closely related *Drosophila* species. For example, cactus can identify as many as orthologs (around 84% to 86%) as orthoMCL in most of the *Drosophila* species. Even in *S. lebanonensis* and other two outgroup species *B. latifrons* and *B. dorsalis*, cactus can identify around 60%-74% of the orthologs in orthoMCL. Whereas MCScanX can only identify around 33% of orthoMCL orthologs in *Bdor*, and the percentages were between 63% and 79% in *Drosophila* species.

The comparison could suggest that cactus may be much better at recovering micro-synteny maps in *Drosophila* genomes. The following tables showed the number of *D. melanogaster* genes that has orthologs identified in orthoMCL, progressive cactus, and MCScanX.

Table. Number of orthoMCL orthologs, aligned orthologs in cactus and orthologs with collinear synteny maps in MCScanX. The number of shared orthologs with orthoMCL identified in Cactus and MCScanX were shown in parentheses.

species	orthoMCL	cactus	MCScanX(Micro-synteny)
Dsim	13486	13584 (10717)	11726 (9870)
Dsec	13427	13624 (11498)	11688 (10582)
Dyak	13229	13429 (10414)	11361 (9245)
Dere	13330	13507 (11439)	11645 (10332)
Dfic	12860	13257 (10868)	11107 (9496)
Drho	12963	13255 (10330)	9241 (7880)
Dele	12860	13242 (10863)	10894 (9416)
Deug	13008	13314 (10874)	11187 (9656)
Dtak	13124	13385 (11048)	10757 (9346)
Dbia	13036	13326 (11087)	11163 (9670)
Dkik	12539	12902 (10642)	10088 (8666)
Dana	12579	12993 (10764)	10538 (9049)
Dper	12138	12592 (10376)	10023 (8427)
Dpse	12174	12639 (10513)	10055 (8565)
Dmir	12193	12635 (10192)	10144 (8250)
Dwil	11795	12205 (9951)	9057 (7449)
Sleb	11483	10401 (8502)	8968 (7455)

Blat	9774	8737 (6822)	4861 (3778)
Bdor	9684	8684 (5785)	4840 (3254)

We also plot the genes that were aligned by progressive cactus or appeared in micro-synteny by MCScanX in closely related species (*Dsim*, *Dsec*, *Dyak*, and *Dere*) and distantly related species (*Sleb*, *Blat*, and *Bdor*). From the two figures attached in the following, we can clearly see that there are more synteny maps (red regions) recovered by progressive cactus alignments than MCScanX with micro-synteny options.

Figure R1. Cactus unique syntenic blocks (red colors) in closely related species.

Cactus unique syntenic blocks (red colors) in distantly related species.

line 92-93: the authors state that their method results in the exclusion of species-specific candidate de novo genes. Why? How? This isn't obvious to me. This seems like an important point to clarify.

Response:

Thanks for bringing up this issue. We have reorganized and rewrote the section *Identification of Drosophilinae lineage-specific de novo gene candidates in D. melanogaster* in *Results* to better describe why there are no *D. melanogaster* specific de novo genes identified in this study.

To clarify here, there are 73 protein coding genes in *D. melanogaster* that were aligned to unannotated syntenic regions in other species, including *D. sim*, and *D. sec*. However, upon further gene structure predictions (*Material and Methods*, section *Identification of de novo gene candidates in D. melanogaster* and section *Random simulations of Genewise/Spaln*), we found that these syntenic regions may harbor putative unannotated orthologs (sometimes partial orthologs) at high confidence in *D. sim* and *D. sec*. Thus, from our analysis, we did not term them as *D. mel* specific. Instead, we identified 23 of them to be possibly *de novo* but placed them on the older branches as there is a likelihood that partial orthologs in *D. sim* or *D. sec* being functional.

The authors analyse G+C content of the candidate de novo genes. They find that de novo genes have lower G+C content. However, statistically speaking there should be more numerous (though shorter) ORFs in G+C poor regions (McLysaght & Hurst, 2016), which might create an ascertainment bias. Have the authors considered this?

Response

We agree that there are many intergenic ORFs with extremely low GC content. To answer this question, we extracted all possible intergenic ORFs from *D. melanogaster* genome and calculated the GC contents of these ORFs. In all, we obtained 372, 675 intergenic ORFs. We found that *de novo* gene candidates have significant higher GC contents than these intergenic ORFs (P-value = $2e-22$). This analysis supports the argument in McLysaght & Hurst, 2016 that AT regions may have strong potentials to produce ORFs (left box plot lower tier). However, the below figure also shows that regions with very low GC (<30%) do not normally become genes. Based on the increases of GC contents from intergenic ORFs to *de novo* gene candidates, and then to other annotated protein coding genes (P-value= $3e-139$), we conclude that this observation is not likely a bias, but rather an underlying properties of *de novo* genes.

Might there also be an influence of lower recombination rates in low G+C regions?

Response

It was suggested that GC content has weak positive correlations with recombination rates. That is, for regions with lower GC, the recombination rates should be smaller.

It has been suggested in yeast that de novo genes tend to be originated in recombination hot spots (Vakirlis et al., 2018, doi: 10.1093/molbev/msx315). To check if similar trend exists in *D. melanogaster*, we obtained regions of different recombination rates in *D. melanogaster* genome as described in Haddrill et al (doi.org/10.1186/gb-2007-8-2-r18) and Charlesworth et al (doi.org/10.1017/S0016672300034029). Although there is a marginal enrichment (hypergeometric test p-value=0.07) in the regions with intermediate recombination rate, we did not observe strong bias in the recombination rates of the de novo gene candidates, suggesting that lower recombination rates in low G+C regions were not likely to have strong bias in *de novo* gene origination.

Combination Rate	de novo genes	other genes	total
No	5 (1%)	144 (1%)	149 (1%)
Low	58 (11%)	1445 (9%)	1503 (11%)
Intermediate	200 (38%)	4310 (27%)	4510 (34%)
High	270 (51%)	6679 (53%)	6949 (53%)
total	533 (100%)	12578 (100%)	13111 (100%)

Could this affect the alignments and the ability to detect these de novo genes?

Response

This is a great point. Thanks for bringing this up. We have checked the alignment coverage, and alignment depth for genes with different GC contents. To do this, we grouped all the genes in to 10 groups according to the ascending values of GC contents. We defined two different metrics to check whether GC contents affects the alignments. The first is alignment depth, where the depth is the furthest branch a *D. mel* gene can be aligned. The second is aligned ratio, where the ratio is the proportions of genes in the

GC group can be aligned in cactus. From the figures attached below, we can see that there are no significant differences in the two metrics examined here among genes in the 10 GC groups.

The authors appear to jump to the conclusion that the GC difference between *de novo* and established genes relates to translational selection (line 142), but I think this is an over-interpretation of scant evidence. This could be explained by an ascertainment bias, so the idea of translational selection would need to be tested before it can be claimed. The same issue comes up on line 188 onwards. I think that exploring the idea of translational selection is interesting, but would need additional testing. For example, is the classic relationship with gene expression level found? Are the codons indeed preferred codons? If the authors are not thinking of translational selection manifesting as codon usage bias then what kind of translational selection are they invoking? How does the gene G+C content relate to that of the surrounding genomic DNA? I think this needs deeper inspection and greater justification and explanation.

Response:

Thanks for bringing up this issue. To distinguish the concept between what we observed and the classic “translational selection”, we have rephrased “translational selection” to “selection on translation” or “selection on codon usage” in main text (line 154-158 and line 207-210 in revised manuscript).

To support this, we calculated proportions of optimal codon usage for each of the amino acids in *de novo* genes and other established genes. We obtained optimal codons for each of the amino acids from the genes that are expressed with TPM greater than 1 in either male or female whole body. We found that except for single codon amino acid, all

the amino acids in de novo genes used less optimized codons significantly than other established genes. The results may indicate that there exists certain selection on translation or codon usage. We have listed the results in Table S1 as following:

Table S1. Proportions of optimal codons in *de novo* genes and other annotated protein-coding genes in *D. melanogaster*. The median values of the proportions were listed in the P(optimal, *de novo*) and P(optimal, other) columns. *De novo* genes show significant less optimal codon usage compared to other annotated protein coding genes. The P-value was computed using *scipy.stats.ttest_ind* module with option *alternative="less"*. was shown in the P(t-test) column. For most of the amino acids, the proportion of optimal codons show significant positive correlation with the origination branches as shown by the P-values of Spearmanr and Kendalltau rank correlation test.

Amino Acid	P(optimal, de novo)	P(optimal, other)	P(t-test)	P(Spearmanr)	P(Kendalltau)
A	0.32	0.45	9.2E-70	4.4E-07	5.3E-07
C	0.62	0.73	3.1E-25	5.9E-02	5.8E-02
D	0.38	0.46	1.3E-16	5.8E-01	5.6E-01
E	0.50	0.68	7.1E-95	6E-07	9.9E-07
F	0.50	0.63	1.6E-18	2.5E-03	2.6E-03
G	0.28	0.42	2.2E-59	2.2E-07	2.2E-07
H	0.50	0.60	5.2E-31	3E-04	3.8E-04
I	0.33	0.48	2.6E-22	1.7E-03	1.9E-03
K	0.53	0.71	1.5E-52	1.2E-01	1.2E-01
L	0.25	0.42	5.6E-94	4.5E-04	4.5E-04
M	1.00	1.00	nan	nan	nan
N	0.50	0.55	5.6E-14	3.7E-01	3.9E-01
P	0.23	0.33	3E-40	2.6E-03	3.1E-03
Q	0.50	0.71	8.2E-71	5.9E-03	5.9E-03
R	0.14	0.30	6.8E-70	5E-07	9.7E-07
S	0.19	0.24	1.5E-18	1.8E-02	1.7E-02
T	0.25	0.38	1.8E-38	5E-05	5.7E-05
V	0.33	0.47	3.9E-70	3.2E-04	3.7E-04
W	1.00	1.00	nan	nan	nan
Y	0.50	0.64	1.3E-09	1.5E-02	1.4E-02

The section on folds and well-folded proteins -- line 205 onwards (page 8 by my reckoning, though the pages are not numbered) lacks some important information in my opinion. "MD simulations" are not explained or introduced. One has to go to the methods to find that MD probably stands for 'molecular dynamics' and even still, the broad methodology/technique is not introduced. What does it really mean to say that the folds remain stable in these simulations? What is being tested here?

Response:

Thanks for bringing up this issue. We have revised this to "molecular dynamics (MD) simulations". In addition, we added an intro here why this technique was used (line 54-60 in revised manuscript). The reason why we did molecular dynamics simulations is because of protein dynamics.

AlphaFold only generated static and rigid low-energy protein structures. AlphaFold does not tell whether the structural fold is stable or not. For example, there are cases where

AlphaFold predicted a relatively short protein encoded by one of the *de novo* gene candidates (FBgn0261949) as a single alpha helix, with averaged PLDDT of 0.94, which falls into the highly confident cases. As predicted by TMHMM2, it is not transmembrane protein. However, a single alpha helix would not be a stable structural fold in physiological conditions nor in MD simulations.

Here through the incorporation of MD simulations, we were able to further assess whether the structural folds predicted by AlphaFold are stable in physiological conditions or not. In some other studies, it was reported that MD simulations could substantially improve the predicted structures by AlphaFold (Schlick et al. 2021, doi.org/10.1038/s43588-021-00060-9; Heo & Feig 2020, doi.org/10.1002/prot.25847).

In the introduction of revised manuscript, we have added an introduction on why we incorporate MD simulations. “Although AlphaFold2 has been proved to be highly accurate, it predicts only single static protein structure per protein sequence (Lane, 2023), which could hinder our understanding on the protein structures of *de novo* gene. Molecular dynamics (MD) simulation has been shown as a valuable tool to investigate protein structural dynamics (Dror et al., 2012), study protein structure stability (Childers & Daggett, 2017), and evaluate or refine predicted or designed protein structures (Heo & Feig, 2020; Schlick & Portillo-Ledesma, 2021). Thus, we further carried out large scale MD simulations to characterize the structural stability and dynamics of the predicted protein structures.”

The authors rely on AlphaFold2 for their structure predictions. My understanding is that this is an *ab initio* (rather than homology modelling) method, so is theoretically capable of predicting structures for proteins with no available homologs. However, I am concerned regarding the interpretation of the differences (or not) between the predicted structure of the modern protein as compared to the inferred ancestral protein sequence (page 10- line 254 onwards). I am guessing that in most cases there are very few substitutions. What power does AlphaFold2 have to possibly return different structures when the sequences are only slightly different?

Response

We agree with the reviewer that AlphaFold2 would not perform well in cases of point mutations. However, *de novo* genes often evolve rapidly, which often resulted in many mutations between the *de novo* genes and their most diverged ancestral forms and the sequence identities between them range from ~30% to ~70%. It's worth noting that, even with sequence identity of 70%, AlphaFold2 can predict the two proteins with lower structure similarity (FBgn0014850, TM-score = 0.59, sequence identity = 71%) than

sequence identity of 30% (FBgn0264746, TM-score = 0.82, sequence identity = 29%). A full list of the table is attached below:

FBID	TM-score(ToDmel)	Sequence Identity
FBgn0037042	0.85961	0.43
FBgn0264748	0.85053	0.36
FBgn0264747	0.89853	0.36
FBgn0014850	0.59064	0.71
FBgn0263647	0.74445	0.43
FBgn0264746	0.818	0.29
FBgn0262819	0.60773	0.61
FBgn0265046	0.73458	0.36
FBgn0260967	0.88649	0.35
FBgn0004593	0.61398	0.56
FBgn0262480	0.49083	0.41
FBgn0261580	0.73576	0.34
FBgn0263250	0.78832	0.58
FBgn0261587	0.66441	0.28
FBgn0265834	0.62028	0.29
FBgn0261581	0.63957	0.32
FBgn0262824	0.70221	0.33
FBgn0262896	0.23644	0.44

On the other hand, to further check the ability of AlphaFold2 in predicting the potentially well-folded protein structures of the de novo genes with limited alignments, we used protein language models (ESMFold) to predict the protein structures where sequence alignments were not necessary. ESMFold is a protein language model that does not rely on multiple sequence alignments. We could see that the structures predicted by AlphaFold2 and ESMFold were highly similar. In the table below, we can see in most cases, the structural models predicted by AlphaFold2 and ESMFold are highly similar with TM-score close to or higher than 0.8.

FBID	TM-score
FBgn0037042	0.9684
FBgn0264748	0.9603
FBgn0264747	0.9399
FBgn0014850	0.4172
FBgn0263647	0.9485
FBgn0264746	0.9460
FBgn0262819	0.7360
FBgn0265046	0.9818
FBgn0260967	0.2512
FBgn0004593	0.2803
FBgn0262480	0.7924
FBgn0261580	0.9858
FBgn0052192	0.8887
FBgn0263250	0.9542
FBgn0261587	0.9109
FBgn0265834	0.9297
FBgn0261581	0.9436
FBgn0262824	0.9149

FBgn0262896	0.5102
-------------	--------

We applied ESMFold again to predict the protein structures of de novo gene ancestral states. For most of the cases, ESMFold predictions were consistent with AlphaFold2 predictions.

Structural similarity (TM-score) and structural discrepancy (RMSD) between AlphaFold2 and ESMFold predictions. For most (~80%) of the cases, AlphaFold2 predictions were highly consistent with ESMFold predictions with TM-score > 0.5 and RMSD < 2Å.

Based on above observations, we are optimistic that AlphaFold2 is able to predict the protein structures at relatively high confidence.

I believe that the method does use alignments in one of the steps, and these alignments will presumably be the same or highly similar for the ancestral sequences. Is it reasonable to expect that AlphaFold2 might be capable of inferring an alternative, or significantly altered structure for the inferred ancestral sequence?

Response

Similar to the point above, we used an independent program, ESMFold, on the ancestral states, similar results were obtained, indicating AlphaFold2 has the potential to study the ancestral protein structures.

We thank the reviewer for bringing up AlphaFold related questions, which pushed us to use ESMFold for validation.

If not, then it is not reasonable to interpret the lack of major differences as reflecting anything of the true biological history of the genes and instead might be a limitation of the method? I am not confident that this is indeed a problem, so if the authors are aware that the method can indeed do what they hope it does, then I think it merits some mention. Either way, I think it would be important to detail the limitations of this approach.

Response

The reviewer is right that there are limitations. We tried to validate AlphaFold2 models with ESMFold and MD simulations. ESMFold uses a language model and is independent of AlphaFold2. Although these methods tend to have high accuracy, we agree with the reviewer that there may still be limitations to infer the results from computational predictions. In the revised manuscript, we added in the end of second paragraph of Discussion that “However, these observations were based on computational predictions.

Further experimental validation would be needed to better understand the protein structures of de novo genes.” (line 406-407 in revised manuscript).

I do not understand the relevance of the MD simulations in this section (line 269) and I don't know what the notation '200 ns' means with respect to the simulations. Perhaps with more information I would understand better and be convinced. At present, I simply find myself wondering whether or not the methods used actually have sufficient scope to infer alternative structures given the underlying sequence similarity. I would appreciate a better explanation of this approach to justify the interpretations.

Response:

Sorry for the confusions. As mentioned above, we wanted to further assess whether the structural folds predicted by AlphaFold are stable in physiological conditions or not. We think that MD simulations could be important especially for ancestral states structural models as to avoid the cases that Alphafold2 could not distinguish homology sequences. In such cases, MD simulations with physics-based molecular force-field could tell “bad” structural models from “good” structural models, as the “bad” structural models would have high potential energies and would be unstable in MD simulations.

The notion “200 ns” means the length of MD simulations, which were conducted for a length of 200 nanoseconds. Previous studies employed various simulation length ranging from ~20 ns to microseconds to study the stability of homology models. Here we used 200 ns to ensure sufficient sampling while making MD simulations feasible in our large scale studies.

We also showed an example above to demonstrate an unstable structure model for FBgn0261849.

The expression analysis (page 11; line 299 onwards) clusters the testis-biased genes into four clusters. However, in k-means clustering the number of clusters is decided in advance. How did the authors decide on making 4 clusters rather than any other number? What is the justification?

Response:

Yes, the number of clusters was decided in advance in k-means clustering. We did perform k-means clustering using different number of clusters and computed the sum of squared errors (SSE). We found that SSE was quite converged with 4 clusters. We added this plot in Figure S4 and the revised Figure S5 is as following. Another justification was from the heatmap in the right column of Figure S5, the expression patterns clearly show 4 clusters.

Figure S5. Clustering of all *D. melanogaster* testis-biased genes. The sum of squared error (SSE) as a function of the number of clusters was shown in the bottom left panel. The genes were finally clustered into 4 clusters and the expression patterns of the 4 clusters were shown in the right column.

Line 309: The authors state that de novo genes in cluster 1 differ from de novo genes in other clusters. However, they don't state whether or not they are similar to other cluster 1 genes (ie non de novo genes with similar expression pattern). What is the basis for the interpretation? What is being tested here?

Response:

This is a great point. We now compared the de novo genes and non de novo genes in cluster 1. We found similar patterns. In cluster 1, *de novo* genes are significantly more disordered and exposed compared to non de novo genes. *De novo* genes show lower but not significant transmembrane and signalp probabilities compared to other genes in cluster 1. For the evolutionary properties, de novo genes in cluster 1 show much higher evolutionary rates, adaptation rates, and higher proportions of adaptive changes compared to other genes. While the nonadaptation rates of de novo genes do not differ with other genes in cluster 1 significantly.

Since de novo genes in cluster 1 showed significantly different properties among testis-biased de novo genes, we hypothesized that these genes could also be important in *de novo* gene origination. We think that although only very few *de novo* genes were enriched in the early spermatogenesis stages, they might also play a non-negligible role in *de novo* gene origination (descriptions in main text can be found in line 337 to 342 in revised manuscript).

The interpretation regarding shifts in pattern of expression with de novo gene age needs greater justification. What might be the biological basis for this?

Response:

This is a great point. It has been shown that new genes or *de novo* genes tend to be in the periphery of cellular networks (Abrusán, 2013). Thus, they were more likely to be tolerated by the host. This provided de novo genes weaker selective constraints for *de novo* genes and the sequence of *de novo* genes could change faster than other conserved genes. The changes could happen in regulatory sequences or coding regions, affecting expression patterns or levels of *de novo* genes. *De novo* genes that stay in the periphery are more likely lost or eliminated due to the weaker selective constraints. While the genes that have different expression patterns may have stronger selective constraints compared to newly originated *de novo* genes, these genes tend to be selected during evolution.

We have added the following discussions in the second last paragraph of Discussion. “Due to the weaker selective constraints, *de novo* genes tend to undergo faster sequence evolution, resulting in abundant sequence changes. These changes could potentially happen in regulatory or coding regions and further affect the expression patterns or expression levels of *de novo* genes. The gradual shift of sequence and expression patterns of *de novo* genes might increase their chances ...” (line 424 to 455 in revised manuscript).

Minor points:

line 90: the term Li is introduced without explanation or expansion (it comes later, in the figure legend, but I think it needs to be in the text too). Furthermore, I think the choice of terminology here is a bit confusing. It seems to be referring to both a branch AND the

clade defined by that branch. I think the term can only be one or the other. As it stands, I found it confusing. I also am unused to seeing the word 'lineage' used to refer to a clade, so that was also a bit confusing.

Response:

Thanks for bring this up. We have rephrased the terminology "lineage" to "branch" to better reflect the species groups in *Drosophila* lineages.

line 99: why was it necessary to remove cases where there have been translocations? Don't these get removed anyway in the later step that considers synteny?

Response:

The reviewer is right that it is not necessary to describe this step here since these genes would be further removed by downstream analysis. We have now rephrased our method to better describe the methods used in our identification pipeline. Please refer to the revised section *Identification of de novo gene candidates in D. melanogaster* in *Material and Methods*.

line 142: I do not understand the interpretation that lower G+C somehow relates to "immature codons" and I don't understand what is meant by that phrase.

Response:

We have rephrased "immature codons" to "unoptimized codons" to reflect what we meant precisely. We have checked the proportions of optimized codon usage for each of the amino acids in each of the de novo genes and compared them to other protein coding genes. Our results show that de novo genes used less optimized codons compared to other protein coding genes, supporting the conclusion that lower GC in de novo genes might be related to unoptimized codons.

line 203: sequential -> sequence

Response: We have revised sequential to sequence (line 222 in revised manuscript).

line 240: I find the section heading to be a bit misleading as only 3/19 have potentially novel folds.

Response: We have rephrased the section heading to "Overall, our results indicated that well-folded de novo genes are likely to adopt existing protein structure folds" to better reflect the contents of this section (line 261 and line 272-273 in revised manuscript).

line 303: "We found that ..." - my understanding is that this isn't a finding, but is the result of the clustering. Rephrase.

Response:

Thank you for your valuable suggestions. We have rephrased the sentence in the revised manuscript as following.

“We numbered the four clusters according to the expression patterns, where genes cluster #1 tend to be highly expressed in early spermatogenesis stage, genes in cluster #2 showed average expression in spermatogonia and spermatocytes stages, genes in cluster #3 showed average expression in spermatocytes and spermatids stages, and genes in cluster #4 showed peak expression in spermatids stage.” (line 326-330 in revised manuscript).

The words unneglectable and non-neglectable are both used in various points in the manuscript. I think the authors perhaps mean non-negligible.

Response: There is a subtle difference among the words. "Negligible" means so small or unimportant that it can be disregarded, while "neglectable" means capable of being neglected on purpose. We have revised the manuscript, removed the word unneglectable, and used non-neglectable and non-negligible to distinguish the two scenarios.

There are various points in the manuscript where there are small syntax errors. In all cases I was confident that I understood the intended meaning, so there was no impediment to understanding, but these should be fixed before final publication. I have not listed them all here, but I do provide a few examples:

line 32: ‘born from scratch through previously non-genic DNA’ should perhaps be ‘born from scratch from previously non-genic DNA’

Response:

Thank you for the suggestion, we have revised “through” to “from” (line 33 in revised manuscript). We have also checked the whole manuscript carefully and by several native English speakers.

line 90-92 - this sentence doesn't make sense to me.

Response:

We have rewrote the paragraph to make it clearer (line 95 to 112 in revised manuscript). “...For each of the 13798 *D. melanogaster* protein-coding genes, we combined homology obtained from all-vs-all blastp (Altschul et al., 1990) analysis, Genewise (Birney et al., 2004) and Spaln (Iwata & Gotoh, 2012) predictions to identify annotated/unannotated orthologs and non-genic hits from their syntenic regions (Figure 1B). For simplicity, we termed the furthest branches that have annotated/unannotated orthologs as Br_i , where i could range from 1 to 9 for each potential candidate, as shown in Figure 1A. The above step gave 1285 potential de novo gene candidates within Br_9 (see Figure 1B and Material and Methods for detail). We then removed genes that have homologs that are not in the syntenic regions using all-vs-all blastp (Altschul et al., 1990). This led to 686 potential de novo gene candidates within Br_9 . As a last filtering step, we removed candidates that have reliable annotated or unannotated homologs outside of Br_9 by blastp and iterative jackhmmer (S. R. Eddy, 2011) search against UniProt Knowledgebase sequence database (UniprotKB) (Bateman et al., 2022) and tblastn search against NCBI representative genomes (Figure 1B, Material and Methods). Finally, combined with homology and synteny, we identified 555 de novo protein-coding gene candidates in *D. melanogaster* that are potentially originated within Drosophilinae

lineage (supplementary File S2: The list of *de novo* gene candidates and their properties).”

line 299 : bad syntax

Response: We updated the sentence to “Of the 555 *de novo* gene candidates identified, many of them (217, ~40%) had biased expression in the testis” (line 322 in revised manuscript).

line 348: taxonomy -> taxonomically

Response: Edited (line 368 in revised manuscript). Thank you.

REVIEWER COMMENTS

Reviewer #1 (Remarks to the Author):

The authors have done a nice job responding to my comments from the initial round of review, and the new manuscript is stronger and clearer as a result. It's now easier to see the novelty and importance of the structural analyses with AlphaFold, and I agree with the authors that this is a significant advance over previous attempts to identify *D. melanogaster* de novo genes. I also appreciate the clearer outline in the text and Fig. 1B of the bioinformatic approaches. I have just a couple remaining questions that relate to genes that had some attributes that might suggest a de novo origin but did not pass the authors' rigorous (and appropriate) analysis pipeline, and then some minor editing suggestions.

Priority points:

1. The new supplemental file showing all examined *D. melanogaster* genes is helpful; thanks for adding this. For the genes that were not called as de novo, the table states that the orthologous locus was not non-genic (i.e., was genic) in the furthest aligned branch. Would it also be possible to add for each gene which branch that was? This would give the reader a better sense of how good the coverage was for each gene across the breadth of species examined in the whole-genome alignments. For those genes for which the syntenic region was genic within *Drosophila* but that could not be aligned outside of *Drosophila*, how are the authors thinking about these? Might they be considered putative de novo genes if the encoded proteins lack detectable homology outside of the genus? Depending on the number of genes that fall into that category, could this be assessed with the types of jackhmmer/tblastn approaches the authors used for some of the other de novo gene candidates? (I appreciate these approaches have some aspects that are not fully automated, so if the number of genes in this category is too many, then it is okay for the authors to opt not to do this. That said, if the number of genes without syntenic regions identified outside of *Drosophila* is large, then it reveals a limitation of the whole genome alignment ortholog finding approach.)

2. For the genes initially called as de novo candidates, but for which tblastn identified a potential homolog in an outgroup species, how frequently were the potential homologous sequences found as parts of annotated genes or represented in previous RNAseq data? I appreciate that the authors also checked these sequences via spaln/genewise, and their argument in the discussion (lines 377-381) about the possibility of recent loss of function in the outgroup species is reasonable, but do how often do independent lines of evidence (e.g., genome annotation, expression data) support these regions being genic?

Minor points:

Line 64: To avoid confusion with proteins involved in cell signaling pathways, "signal proteins" should be changed to "containing a signal peptide." There's a similar issue in line 351, where proteins are described as "signal."

Line 65: Do you mean "the structure of de novo proteins" instead of "the structure of de novo genes"? The above paragraph mostly describes protein structural features, though gene structure is

also an interesting question in the field. (Line 71 also refers to de novo genesTM structures, but might mean the structures of the encoded proteins.)

Line 67: "ranging from very young and old" is awkward, perhaps "of varying ages" instead?

Fig. 4B still uses L_ to indicate gene age/phylogenetic branch instead of the new Br_ notation. The same issue recurs in the lineage_age column in the supplemental file listing all the de novo and in the file showing the genes that were excluded due to blastp/jackhammer/tblastn searches. This latter file also uses L10 in the tblastn section; does this refer to a specific branch, or to any branch more basal than L/Br9?

In the fourth box up from the bottom in Fig. 1B, "are" can be removed from "Identify D. mel genes that are have orthologs";

Line 297: "undergo" should be "have undergone"; lines 298-299: "their"  "its"

Line 446/456: define PPI when it is first used (at line 446) instead of at 456

Reviewer #3 (Remarks to the Author):

Overall, this manuscript provides a valuable addition to the study of de novo gene evolution on multiple levels. It presents a practical and adaptable method for identifying potential orthologs of de novo genes in a focal species, incorporating both BLAST and synteny-based techniques. A notable innovation is the integration of whole-genome alignments in this study (cactus), which appears to be advantageous in discerning the non-coding sequences in outgroup species from which these genes might have originated. This approach significantly enhances the likelihood of successfully identifying de novo genes in numerous instances. Additionally, the authors investigated the potential foldability of de novo proteins and their ancestors using modern structure predictors and MD simulations. These MD simulations reveal that for many de novo proteins high pLDDT and globular structures are provided by the structure predictors, while the predicted structures are not stable during MD simulations.

The comments and suggestions of Reviewer 2 have been sufficiently incorporated into the manuscript. Nevertheless, there remain certain issues that have not been raised by the two other reviewers.

We should emphasize that addressing these issues would mainly help increase the reliability of the data and we do not question the principal validity of the results.

MAJOR points:

Regarding the structure and disorder predictions: The authors do not clearly state which disorder predictor they use and they cite the CAID results from 2021 with Necci et al., 2021. According, to CAID fIDPnn is the best disorder predictor and has been shown in Liu et al., Proteins (2023) and Aubel et al., F1000research (2023) to be the most applicable disorder predictor for de novo proteins. This should be clearly stated in the method section.

The MD simulations will have to be performed in triplicates. A single run of an MD simulation is not sufficient.

Also, it must be clear that the burn-in phase has been repeatedly overcome.

Regarding the structure prediction of ancestrally inferred sequences, Alphafold would pick up on the same sequences in its MSA generation from which the ancestral sequence was created from. This would create a bias which the authors could circumvent using ESMfold additionally as they have done before but should be in general mentioned.

It is not fully clear what is meant with P(confident) in Figure 3?

MINOR points:

In general, the pitfalls of structure prediction of de novo proteins or singletons have been recently discussed in Monzon et al., 2022, Aibel et al., 2023, Middendorf & Eicholt, bioarxiv, 2023 and Liu et al., 2023 and might further support and explain the results of the authors here

line 264: Foldseek would be a more modern choice over RUPEE but that is the authors decision to make

The bias of Alphafold predicting high-confidence structures that are actually not stable in MD might be sequence-length dependent since the AMBER force field based energy minimization in the final structure module of Alphafold is likely to force smaller proteins into an unrealistic low energy conformation (see Monzon et al., 2022, Eicholt et al., 2022, Middendorf & Eicholt, 2023). This could also be tested through shuffle or proline insertion into these smaller proteins and predicting them once more. Likely similar predicted structures would come out, while being biophysically impossible even observable by eye on the sequence. Therefore, we would be cautious with the claims in lines 397-389

line 479-481: One technical discrepancy between these studies is the different use of lupred, discussed in Aibel et al., 2023.

line 144-145: Please provide a citation for this claim that flies have lower number of TEs than humans

line 495: Vakirlis instead of Vikrilis

REMARKS:

Could the authors please explain (possibly in the the supplemental methods) how the cactus alignments help obtain more reliable results than a case-by-case synteny approach using plastp for the known genes which can serve as "anchors".

The latter has been frequently used in several publications by looking up the closest up- and down-stream neighbours or a putative de novo gene

with a (possibly) spurious hit. This question comes up considering the loss of gene order which has been demonstrated by Zdobnov+Bork and makes one wonder how good just any cactus whole-genome alignment can be in the case of fly genomes and if it is of any help after all.

Accordingly, while newly introduced Figure R1 looks great, it would be assumed that requiring conserved gene-micro-synteny could be more stringent. Overall, the pipeline is great and it is impressive it also identifies lineage specific duplicates (paralogs) of de novo genes.

Just for clarification: does it also identify duplicates of genes which are clearly de novo but with both copies remaining present over one (or more) speciation event?

This case seems to be not too rare considering recent results from Grandchamp et al. 2023, Genome Research.

(We agree this is a side-topic and need not be further pursued technically, again a brief explanation in supplement would help).

When stating that "Overall, our results indicate that well-folded de novo genes are likely to adopt protein structure folds"

(btw please not genes do not fold): are you suggesting that the vast majority of all de-novo proteins fold convergently?

This would be difficult to align with the 4 de novo proteins which have been (incompletely) structure analysed and the common knowledge of protein folding.

Not that we should not overturn current dogmata if need be, but how accurate is that statement considering length of de novo proteins and the definition of a fold?

Response to reviewers' comments

Reviewer #1 (Remarks to the Author):

The authors have done a nice job responding to my comments from the initial round of review, and the new manuscript is stronger and clearer as a result. It's now easier to see the novelty and importance of the structural analyses with AlphaFold, and I agree with the authors that this is a significant advance over previous attempts to identify *D. melanogaster* de novo genes. I also appreciate the clearer outline in the text and Fig. 1B of the bioinformatic approaches. I have just a couple remaining questions that relate to genes that had some attributes that might suggest a de novo origin but did not pass the authors' rigorous (and appropriate) analysis pipeline, and then some minor editing suggestions.

Response:

Thank you very much for recognizing the improvements/revisions we made to the initial manuscript. We have now further revised the manuscript according to your valuable suggestions. We thank the reviewer again for the thoughtful comments.

Priority points:

1. The new supplemental file showing all examined *D. melanogaster* genes is helpful; thanks for adding this. For the genes that were not called as de novo, the table states that the orthologous locus was not non-genic (i.e., was genic) in the furthest aligned branch. Would it also be possible to add for each gene which branch that was? This would give the reader a better sense of how good the coverage was for each gene across the breadth of species examined in the whole-genome alignments. For those genes for which the syntenic region was genic within *Drosophila* but that could not be aligned outside of *Drosophila*, how are the authors thinking about these? Might they be considered putative de novo genes if the encoded proteins lack detectable homology outside of the genus? Depending on the number of genes that fall into that category, could this be assessed with the types of jackhmmer/tlbastrn approaches the authors used for some of the other de novo gene candidates? (I appreciate these approaches have some aspects that are not fully automated, so if the number of genes in this category is too many, then it is okay for the authors to opt not to do this. That said, if the number of genes without syntenic regions identified outside of *Drosophila* is large, then it reveals a limitation of the whole genome alignment ortholog finding approach.)

Response:

Thank you for bringing up this question. We did not include lineage-specific genes that could not be aligned in Cactus whole-genome alignments and lack homologs outside of *Drosophila*. The rationale behind this that, without the support of nongenic homologous/orthologous DNA sequence, these genes could potentially undergo different alternative mechanisms other than de novo origination, such as divergence after gene duplication or horizontal gene transfer. Given the long divergence time, we decided to be conservative and not include these as de novo genes in our list.

To answer the question of how many such lineage-specific genes would be putative de novo genes, we further checked our data and found there are 444 potential *Drosophila* specific genes without Cactus alignments in their outgroup species or outside *Drosophila* species. By further

jackhammer and tblastn filtering, we found that 385 of them have no annotated or inferred homologs. If jackhammer and tblastn results represent comparisons to the full complexity of proteomes in nature, one could argue that a subset of these 385 (which is a large proportion of the 444) genes are potentially de novo. However, as explained above, these genes are not included in de novo analysis due to low levels of evidence. Following the reviewer's suggestion, we have labeled the 385 lineage-specific genes and their furthest aligned branches obtained from Cactus whole genome alignments in the revised supplementary file S5.

2. For the genes initially called as de novo candidates, but for which tblastn identified a potential homolog in an outgroup species, how frequently were the potential homologous sequences found as parts of annotated genes or represented in previous RNAseq data? I appreciate that the authors also checked these sequences via spaln/genewise, and their argument in the discussion (lines 377-381) about the possibility of recent loss of function in the outgroup species is reasonable, but do how often do independent lines of evidence (e.g., genome annotation, expression data) support these regions being genic?

Response:

There were 34 de novo candidates removed in this filtering step. Thanks again for the suggestion. We found their potential homologous sequences with significant tblastn E-value and spaln/genewise predictions in a handful of species (12 out of the thousands of species used, mostly *Drosophila* or insect species). These potential homologs are not annotated (if annotated, we would have captured this in our previous pipeline). To answer the question of how often they are expressed, we were able to download RNA-sequencing data for 5 species that harbor putative homologs for these 34 candidates from NCBI. The five species cover 25 of the 34 filtered de novo candidates.

We added the predicted homologous coding sequence to all the coding sequences of the corresponding genomes and used kallisto (Bray et al, *Nature Biotechnology* 34, 525–527 (2016), doi:10.1038/nbt.3519) to estimate the transcription abundance of the predicted coding sequence. For example, if the predicted homologous sequence was found in *Scaptodrosophila*, we added the predicted coding sequence from spaln/genewise to all the coding sequences of *Scaptodrosophila*. We then used kallisto, a near-optimal RNA-Seq quantification tool (doi.org/10.1038/nbt.3519), to map the raw reads and estimate TPM of the predicted coding sequence. Among the homologous sequences of the 25 now-removed de novo candidates, we found 19 of them have estimated TPM greater than 1, indicating a high possibility that these genes are expressed. This suggests that searching genomes and transcriptomes using tblastn/blast is useful in removing possible artifacts, even though these artifacts only account for a few percent of the total candidates.

Minor points:

Line 64: To avoid confusion with proteins involved in cell signaling pathways, “signal proteins” should be changed to “containing a signal peptide”. There's a similar issue in line 351, where proteins are described as signal.

Response:

We have changed “signal proteins” to “containing a signal peptide” accordingly (lines 57, 172, 338 and 620 in revised manuscript).

Line 65: Do you mean the structure of de novo proteins instead of the structure of de novo genes? The above paragraph mostly describes protein structural features, though gene structure is also an interesting question in the field. (Line 71 also refers to de novo genes' structures, but might mean the structures of the encoded proteins.)

Response:

Yes, we refer to the protein structures of *de novo* genes. To reflect this idea, we have rephrased "structure" to "protein structures" accordingly (Line 58 and 63 as well as other protein structures in revised manuscript).

Line 67: "ranging from very young and old" is awkward, perhaps "of varying ages" instead?

Response:

Thanks for the suggestion, we have rephrased the sentence accordingly. Now the sentence is: "To address this question, it is necessary to identify and compare branch-specific *de novo* genes of varying ages within a relatively diverged lineage."

Fig. 4B still uses L_ to indicate gene age/phylogenetic branch instead of the new Br_ notation. The same issue recurs in the lineage_age column in the supplemental file listing all the de novo and in the file showing the genes that were excluded due to blastp/jackhammer/tblastn searches.

Response:

We have now changed them to Br_ notations.

For Fig. 4B, we have also carried out another two replicates of MD simulations and updated the figure.

This latter file also uses L10 in the tblastn section; does this refer to a specific branch, or to any branch more basal than L/Br9?

Response:

We are sorry for the confusions. L10 does not refer to a specific branch, but any branches more distant than Br9, we have now changed "L10" to "more distant than Br9".

In the fourth box up from the bottom in Fig. 1B, "are" can be removed from "Identify D. mel genes that are have orthologs"

Response:

Thanks for correcting the typo. We have removed “are” in revised Figure 1.

Line 297: “undergo” should be “have undergone”. lines 298-299: “their” → “its”

Response:

We have corrected the typo accordingly (line 285 to 287 in revised manuscript)

Line 446/456: define PPI when it is first used (at line 446) instead of at 456

Response:

We now have used “protein-protein interactions” when it is first used (line 437 in revised manuscript).

Reviewer #3 (Remarks to the Author):

Overall, this manuscript provides a valuable addition to the study of de novo gene evolution on multiple levels. It presents a practical and adaptable method for identifying potential orthologs of de novo genes in a focal species, incorporating both BLAST and synteny-based techniques. A notable innovation is the integration of whole-genome alignments in this study (cactus), which appears to be advantageous in discerning the non-coding sequences in outgroup species from which these genes might have originated. This approach significantly enhances the likelihood of successfully identifying de novo genes in numerous instances. Additionally, the authors investigated the potential foldability of de novo proteins and their ancestors using modern structure predictors and MD simulations. These MD simulations reveal that for many de novo proteins high pLDDT and globular structures are provided by the structure predictors, while the predicted structures are not stable during MD simulations.

The comments and suggestions of Reviewer 2 have been sufficiently incorporated into the manuscript. Nevertheless, there remain certain issues that have not been raised by the two other reviewers.

We should emphasize that addressing these issues would mainly help increase the reliability of the data and we do not question the principal validity of the results.

Response

Thank you for your valuable evaluation and constructive comments. We have conducted two additional replicates of MD simulations. We have also included additional discussions on the current limitations of AlphaFold2 and whole-genome alignments. We hope the reviewer finds our response and revision adequate.

MAJOR points:

Regarding the structure and disorder predictions: The authors do not clearly state which disorder predictor they use and they cite the CAID results from 2021 with Necci et al., 2021. According to

CAID, fIDPnn is the best disorder predictor and has been shown in Liu et al., Proteins (2023) and Aubel et al., F1000research (2023) to be the most applicable disorder predictor for de novo proteins. This should be clearly stated in the method section.

Response

Thanks for the thoughtful comments. We used AUCPreD in our manuscript and now we have added a reference to AUCPreD in the revised manuscript. As the reviewer suggested that fIDPnn has been the best disorder predictor as shown in Liu et al 2023 and Aubel et al 2023. We are thankful for their insight, and wanted to note that the predictions in this study were carried out by AUCPreD before the benchmark of fIDPnn (Aubel et al 2023) was available. According to the CAID (critical assessment of protein intrinsic disorder prediction), AUCPreD, together with fIDPnn, was reported to be one of the top five predictors as reported in the 2021 CAID results.

To check if this also applied to *de novo* proteins, we used fIDPnn and another two recently developed structural disorder predictors, ADOPT (Redl et al 2023, doi: 10.1093/nargab/lqad041, deep learning predictor based on the protein language model esm-1b, which has been successfully applied in ESMFold) and AlphaFold_disorder (Piovesan et al 2022, doi: 10.1002/pro.4466, which integrate AlphaFold PLDDT and the solvent accessibility of AlphaFold predicted models and showed great enhancement over PLDDT metric alone).

We observed that AUCPreD highly correlated with other predictors. The Pearson correlation coefficient and P-values are as following:

AUCPreD – fIDPnn: $r = 0.58$, $p = 1e-50$
AUCPreD – ADOPT: $r = -0.46$, $p = 3e-30$
AUCPreD – AlphaFold_disorder: $r = 0.47$, $p = 3e-32$

We further correlated the structural disorder results with the *de novo* gene ages. Similar to the results from AUCPreD, we observed very weak correlations with extremely small correlation coefficients as follows,

Predictors	Kendall tau rank correlation	Spearmanr rank correlation	Pearsonr correlation
AUCPreD	$r = -0.03$, $p = 0.32$	$r = -0.04$, $p = 0.32$	$r = -0.01$, $p = 0.81$
fIDPnn	$r = -0.05$, $p = 0.02$	$r = -0.07$, $p = 0.02$	$r = -0.03$, $p = 0.27$
ADOPT	$r = -0.03$, $p = 0.15$	$r = -0.04$, $p = 0.15$	$r = -0.01$, $p = 0.71$
AlphaFold_disorder	$r = -0.01$, $p = 0.29$	$r = -0.01$, $p = 0.28$	$r = -0.01$, $p = 0.74$

The correlations further supported our conclusion that structural disorder of de novo proteins changed little with their origination ages, as various state-of-the-art structural disorder predictors gave similar results. We have added a revised Figure S4 to include this important result, which is shown as follows for the reviewer's reference.

Figure S4. Structural disorder of de novo proteins by different state-of-the-art predictors, including AUCPreD (top left panel), fIDPnn (top right panel), ADOPT (bottom left panel), and AlphaFold_disorder (bottom right panel). The results overall indicated that the structural disorder of de novo proteins changed little with their origination ages. In addition, we have modified the methods part accordingly. Now the revised section is as follows,

“For protein property predictions, we used deepcnf (S. Wang, Li, et al., 2016) to predict per residue probability of helix, sheet, coil, and solvent accessibility, AUCPreD (S. Wang, Ma, et al., 2016) to predict structural disorder, and PredMP (S. Wang et al., 2019) to predict transmembrane probability. These properties were further normalized by protein length. These structural property predictors have been shown to have high accuracy compared to other methods (Necci et al., 2021; S. Wang et al., 2019; S. Wang, Li, et al., 2016; Y. Yang et al., 2016). For example, in the critical assessment of protein intrinsic disorder prediction by Necci et al, the authors found that AUCPreD, along with fIDPnn (Hu et al., 2021), were consistently among the top five predictors (Necci et al., 2021). To further rule out the bias from the structural disorder predictors, we further used fIDPnn, language model-based predictor ADOPT (Redl et al., 2023), and AlphaFold derived predictor AlphaFold_disorder (Piovesan et al., 2022), to predict the structural disorder for *de novo* proteins.”

The MD simulations will have to be performed in triplicates. A single run of an MD simulation is not sufficient.

Also, it must be clear that the burn-in phase has been repeatedly overcome.

Response

Thank you for the valuable suggestion. We have conducted two additional 200 ns replicates for each of the potentially well-folded protein structures.

Now we have three 200 ns MD replicates for each of the potentially well-folded de novo proteins as well as their ancestral states. We recorded the trajectories every 100 ps. We then performed density peaks clustering on the MD ensemble of each de novo protein and computed the pairwise

TM-score of the representative conformations. To overcome sampling bias of individual replicates, the MD ensemble were constructed by extracting and combining the last 100 ns trajectories (to remove the first 100 ns burn-in phase) from all the 3 replicates.

For many of the de novo proteins, the pairwise TM-scores were similar to previous single simulations, suggesting these structures remained highly stable in replicate simulations. While in some cases, we observed slightly decrease in pairwise TM-scores. Note that the TM-score all remained close to and above 0.7 (column TM-score, 3reps, highlighted in red), and RMSD of the core regions all close to or below 3 Å (column RMSD Core, 3reps, highlighted in red), suggesting the overall structural fold remained stable while some parts of the structures were flexible.

FBID	Name	RMSD FL (single)	RMSD FL (3reps)	RMSD CORE (single)	RMSD CORE (3reps)	TM-score (single)	TM-score (3Reps)
FBgn0037042	CG12984	1.24	1.949	1.24	1.876	0.95	0.91
FBgn0264748	CG44006	1.92	1.727	1.41	1.672	0.96	0.95
FBgn0264747	CG44005	1.46	2.225	1.37	1.87	0.97	0.94
FBgn0014850	Eig71Ej	1.55	4.528	1.51	1.902	0.87	0.76
FBgn0263647	CG43638	1.06	1.218	1.06	1.184	0.95	0.94
FBgn0264746	CG44004	1.49	2.58	1.29	1.744	0.97	0.94
FBgn0262819	CG43190	1.04	2.873	1.04	2.082	0.95	0.83
FBgn0265046	CG44163	1.17	1.704	1.12	1.268	0.94	0.92
FBgn0260967	CG42590	0.99	1.472	0.99	1.472	0.98	0.96
FBgn0004593	Eig71Ef	1.77	5.552	1.30	2.507	0.89	0.69
FBgn0262480	CG43070	0.72	1.411	0.72	1.243	0.97	0.92
FBgn0261580	CG42690	2.57	3.673	1.89	2.68	0.85	0.77
FBgn0052192	CG32192	1.99	3.982	1.60	2.757	0.91	0.76
FBgn0263250	CG43393	0.82	2.092	0.82	1.693	0.97	0.87
FBgn0261587	CG42697	1.14	2.291	0.99	2.071	0.96	0.86
FBgn0265834	CG44623	2.55	4.331	1.60	2.374	0.91	0.79
FBgn0261581	CG42691	1.67	4.94	1.50	3.071	0.91	0.71
FBgn0262824	CG43195	1.40	3.331	1.36	2.388	0.92	0.81
FBgn0262896	CG43251	1.28	2.843	1.02	1.849	0.81	0.68

We noticed similar results for the ancestral structures. For many of the cases, TM-score changed little and in some cases TM-score slightly decreased. In the case of CG43251 where ancestral structures were proposed to be disordered in the current study, TM-score from 3 MD replicates showed a much lower value compared to single MD simulations. After checking the 3 MD replicates, we found that this is due to more sufficient sampling compared to single MD simulations. In single MD simulations, disordered proteins could be trapped in energy minimums,

which is common in MD simulations of disordered proteins (See Kasahara et al 2019, doi: 10.1016/j.csbj.2019.06.009).

In all, the additional MD replications did not affect our conclusions. Rather, with the reviewer’s suggestion, we managed to better characterize the structural dynamics of the predicted structures of the *de novo* proteins. We have updated the results in the revised supplementary_S3_md simulations_ancestral_states. We thank the reviewers again for their insight.

Regarding the structure prediction of ancestrally inferred sequences, AlphaFold would pick up on the same sequences in its MSA generation from which the ancestral sequence was created from. This would create a bias which the authors could circumvent using ESMfold additionally as they have done before but should be in general mentioned.

Response

We now have mentioned that, except for AlphaFold2, we also used ESMfold to predict ancestral protein structures (see line 666 in revised manuscript)

It is not fully clear what is meant with P(confident) in Figure 3?

Response

P(confident) is the percentage of confident AlphaFold2 predictions, which means the percentage of residues that have pLDDT greater than 70. We have now rephrased the figure captions in Figure 3B, which is attached below for the reviewer’s reference. In addition, to better reflect the fact that PLDDT metric is often in the range of 0 to 100 rather than 0 to 1, we have revised the axis labels in Figure 3A-B accordingly. Similarly, in Figure 3A, we have scaled pairwise TM-score from trRosetta predictions by 100 so that the results are comparable with AlphaFold2, ESMFold, and RoseTTAFold.

Figure caption of Figure 3B, “Most *de novo* gene candidates might not have the well-folded protein structures according to pLDDT, per-residue confidence score, and P(confident), which represents the percent of residues that were confidently predicted by AlphaFold2 with pLDDT greater than 70.”

MINOR points:

In general, the pitfalls of structure prediction of *de novo* proteins or singletons have been recently discussed in Monzon et al., 2022, Aubel et al., 2023, Middendorf & Eicholt, bioarxiv, 2023 and Liu et al., 2023 and might further support and explain the results of the authors here.

Response

We are grateful for the reviewer to provide additional information to support and explain our results. We apologize that we could not find publications regarding Liu et al. 2023. In the revised manuscript, we further discuss the pitfalls of structure predictions of *de novo* proteins or singletons and cited the mentioned publications as follows (line 393 to 398 in revised manuscript):

“Note that, recent studies have shown that AlphaFold2 may fold some small lineage specific or *de novo* proteins into unrealistic simple low energy conformations (Aubel et al., 2023; Middendorf & Eicholt, 2023; Monzon et al., 2022). In our study, we applied different deep learning predictors (AlphaFold2 and ESMFold) as well as MD simulations to partially overcome this limitation. However, since these observations were based on computational predictions, further experimental validation is needed to better understand the protein structures of *de novo* genes.”

line 264: Foldseek would be a more modern choice over RUPEE but that is the authors decision to make

Response

Thanks for the valuable suggestion. We will keep in mind and use Foldseek in future studies.

The bias of Alphafold predicting high-confidence structures that are actually not stable in MD might be sequence-length dependent since the AMBER force field based energy minimization in the final structure module of Alphafold is likely to force smaller proteins into an unrealistic low energy conformation (see Monzon et al., 2022, Eicholt et al., 2022, Middendorf & Eicholt, 2023). This could also be tested through shuffle or proline insertion into these smaller proteins and predicting them once more. Likely similar predicted structures would come out, while being biophysically impossible even observable by eye on the sequence. Therefore, we would be cautious with the claims in lines 397-389

Response

The reviewer is right that the claims should be rephrased to better reflect the results from Tejero et al., 2022. We have now rephrased it to (line 384-386 in revised manuscript),

“It was also reported that AlphaFold2 predictions were comparable to high-resolution structure determination techniques, such as solution NMR and x-ray crystallography, especially for some small and relatively rigid single-domain proteins (Tejero et al., 2022)”

In addition, we added a limitation of AlphaFold2 and cited Monzon et al., 2022, Eicholt et al., 2022, and Middendorf & Eicholt, 2023 as follows (also discussed in the response above),

“Note that, recent studies have shown that AlphaFold2 may fold some small lineage specific or *de novo* proteins into unrealistic simple low energy conformations (Aubel et al., 2023; Middendorf & Eicholt, 2023; Monzon et al., 2022). In our study, we applied two different deep learning predictors (AlphaFold2 and ESMFold) as well as MD simulations to partially overcome this limitation. However, since these observations were based on computational predictions, further experimental validation is needed to better understand the protein structures of *de novo* genes.”

line 479-481: One technical discrepancy between these studies is the different use of Iupred, discussed in Aubel et al., 2023.

Response.

We have added a discussion and cited Aubel et al, 2023 accordingly in the revised manuscript (line 468-469 in revised manuscript). The discussion is as follows,

“For example, a recent study revealed that the choice of protein structural disorder predictors could result in discrepancies (Aubel et al., 2023).”

line 144-145: Please provide a citation for this claim that flies have lower number of TEs than humans

Response

We have now cited Yang et al 2022 (doi: 10.1371/journal.pgen.1010024) (line 134 in revised manuscript). In this reference, the authors described “This TE burden can range from the extreme >70% proportion of the axolotl genome to >50% in the human genome to >10% in the *Drosophila melanogaster* genome.”

line 495: Vakirlis instead of Vikrilis

Response

We have corrected the typo (line 484 in revised manuscript).

REMARKS:

Could the authors please explain (possibly in the supplemental methods) how the cactus alignments help obtain more reliable results than a case-by-case synteny approach using plastp for the known genes which can serve as "anchors".

The latter has been frequently used in several publications by looking up the closest up- and down- stream neighbours or a putative de novo gene with a (possibly) spurious hit. This questions comes up considering the loss of gene order which has been demonstrated by Zdobnov+Bork and makes one wonder how good just any cactus whole-genome alignment can be in the case of fly genomes and if it is of any help after all.

Accordingly, while newly introduced Figure R1 looks great, it would be assumed that requiring conserved gene-micro-synteny could be more stringent. Overall, the pipeline is great and it is impressive it also identifies lineage specific duplicates (paralogs) of de novo genes.

Response

Thanks for bringing up this issue. As the reviewer pointed out, synteny approaches used *blastp* of known protein-coding genes as anchors. Thus, synteny approaches would likely be limited by 1) the quality of available annotations and 2) the availability of close-enough protein coding genes.

Our results show that whole genome alignment approaches, such as Cactus, could supplement synteny approaches by overcoming the two limitations. Cactus determines “anchors” automatically, where each anchor in cactus alignment was maximal gapless contiguous aligned sequences (Paten et al. 2011, doi 10.1101/gr.123356.111) determined by iterative pairwise alignments, filtering of best local alignments, and the construction of filtered cactus graphs.

Different from gene-micro-synteny methods, these best aligned sequences or filtered “anchors” include not only conserved protein-coding genes, but also non-coding regions, such as lncRNA genes or other conserved regions. Thus, the synteny blocks obtained from cactus does not rely on annotations or nearby protein-coding genes. This could potentially result in more “anchors” and more synteny blocks than synteny approaches, and thus less affected by the loss of gene orders.

To better explain the potential advantage of cactus whole genome alignment, we have now added Figure S8 (the previous response Figure), and Table S4 (the previous response Table), where we found that cactus aligner could better recover synteny blocks and orthologs, respectively.

Accordingly, we have added a section “*Micro-synteny and orthoMCL analysis*” in *Material and Methods*.

For the reviewer’s reference, we have attached Figure S8 and Table S4 as follows,

Figure S8. Comparison of synteny blocks recovered by cactus aligner and micro-synteny method in (A) four closely related genomes (*D. simulans*, *D. sechellia*, *D. yakuba*, and *D. erecta*), and (B) three distantly related genomes (*S. lebanonensis*, *B. dorsalis*, and *B. latifrons*). The figure shows that Cactus aligner recover more syntenic regions.

Table S4. Number of *D. melanogaster* protein-coding gene orthologs recovered by orthoMCL, Cactus aligner, and MCscanX with micro-synteny option. The number of overlaps between Cactus and orthoMCL, and MCScanX and orthoMCL are shown in parenthesis.

Species	orthoMCL	Cactus	MCScanX(Micro-synteny)
Dsim	13486	13584 (10717)	11726 (9870)
Dsec	13427	13624 (11498)	11688 (10582)

Dyak	13229	13429 (10414)	11361 (9245)
Dere	13330	13507 (11439)	11645 (10332)
Dfic	12860	13257 (10868)	11107 (9496)
Drho	12963	13255 (10330)	9241 (7880)
Dele	12860	13242 (10863)	10894 (9416)
Deug	13008	13314 (10874)	11187 (9656)
Dtak	13124	13385 (11048)	10757 (9346)
Dbia	13036	13326 (11087)	11163 (9670)
Dkik	12539	12902 (10642)	10088 (8666)
Dana	12579	12993 (10764)	10538 (9049)
Dper	12138	12592 (10376)	10023 (8427)
Dpse	12174	12639 (10513)	10055 (8565)
Dmir	12193	12635 (10192)	10144 (8250)
Dwil	11795	12205 (9951)	9057 (7449)
Sleb	11483	10401 (8502)	8968 (7455)
Blat	9774	8737 (6822)	4861 (3778)
Bdor	9684	8684 (5785)	4840 (3254)

Just for clarification: does it also identify duplicates of genes which are clearly de novo but with both copies remaining present over one (or more) speciation event?

This case seems to be not too rare considering recent results from Grandchamp et al. 2023, Genome Research.

(We agree this is a side-topic and need not be further pursued technically, again a brief explanation in supplement would help).

Response

The pipeline could potentially identify branch specific duplicates. For example, the two genes, FBgn0051909 and FBgn0264344, were identified as duplicated *de novo* gene candidates originated in Branch 1. In cactus whole genome alignments, the two paralogs were *D. melanogaster* specific, with their Dsim ortholog to be Dsim|GD23456, and Dsec ortholog Dsec|LOC6611512.

In another example, FBgn0029694 and FBgn0037910, they appeared to have two copies in Dsim and Dsec, three copies in Dyak and Dere, but only one copy in Dfic and Drho, suggesting a possible duplication event during the speciation of melanogaster subgroup species.

Since our main topic in current study is on the origin and structural evolution of de novo genes, how the *de novo* genes changed their gene copy numbers after origination would be beyond the scope of our current manuscript. We thank the reviewer for bringing up this issue and for understanding that this would be an interesting side topic to study in the future.

When stating that "Overall, our results indicate that well-folded de novo genes are likely to adopt protein structure folds"

(btw please not genes do not fold): are you suggesting that the vast majority of all de-novo proteins fold convergently?

Response

Sorry for the confusions. For this conclusion, we only referred to 16 out of the 19 potentially well-folded *de novo* proteins examined. To be a potentially well-folded protein, we required that its AlphaFold2 prediction has: 1) average per-residue confidence score (pLDDT) greater than 80, and 2) the fraction of confidently predicted residues (pLDDT > 70) greater than 90%. We haven't examined other *Drosophila de novo* genes, nor did we study *de novo* genes outside of *Drosophila*. To avoid the confusion, we have rephrased the sentence to "Overall, our results indicated that many of the potentially well-folded de novo proteins examined in our study are likely to adopt existing protein structure folds." (line 260-261 in revised manuscript)

This would be difficult to align with the 4 de novo proteins which have been (incompletely) structure analysed and the common knowledge of protein folding. Not that we should not overturn current dogmata if need be, but how accurate is that statement considering length of de novo proteins and the definition of a fold?

Response

To avoid the confusion, the conclusion was rephrased to reflect the results as discussed in the response to the above comment.

The conclusion was made on *Drosophila de novo gene* candidates with potentially well-folded protein structures. The 4 *de novo* proteins which have been previously (incompletely) structure analyzed did not fall into this potentially well-folded category (discussed in above response), as they have some parts being disordered or flexible.

For the 19 potentially well-folded *de novo* proteins examined in our study, we used TM-score, one of the most widely used metric (others are GDT, RMSD, etc.), to define protein structure similarity. TM-score was designed to be length independent, as discussed in an earlier paper by Xu et al. 2010 (doi: 10.1093/bioinformatics/btq066).

REVIEWERS' COMMENTS

Reviewer #1 (Remarks to the Author):

I appreciate the authors' careful attention to my questions and suggestions from the previous round of review. I find their responses and edits to be satisfactory and think the manuscript is now suitable for publication. I congratulate the authors on a nice study that will be quite influential in the broader field of de novo gene evolution, and also of use to groups working specifically in *Drosophila*.

Reviewer #3 (Remarks to the Author):

The authors have implemented all our comments and suggestions.